# Less Data, Faster Training
## repeating smaller datasets speeds up learning via sampling biases

**Jingwen Liu** [1]  **Ezra Edelman** [2]  **Surbhi Goel** [2]  **Bingbin Liu** [3]

## Abstract

This work investigates the "small-vs-large gap", where repeating on *fewer samples* can lead to *compute saving* during training compared to using a larger dataset. This is observed across algorithmic tasks, architectures and optimizers and cannot be explained using prior theory. We argue that the speedup comes from appropriate layer-wise growth enabled by *sampling biases*, which is more pronounced when the dataset size is smaller. We provide both theoretical analysis and empirical evidence from various interventions. Our results suggest that using a smaller dataset with more repetitions is not just a fallback strategy under data scarcity, but can be proactively leveraged as a favorable inductive biases for optimization, particularly in reasoning tasks.

## 1 Introduction

The conventional wisdom on data use is the more the better, a view supported by both classic generalization theory and extensive empirical evidence [13, 21]. Recent work has reported a counterintuitive phenomenon that fewer samples can lead to *faster* learning. One example is online SGD for single-index models, where taking more than one gradient steps on the same batch can lead to faster convergence in terms of steps [9, 3, 19]. Similarly, empirical study on Transformers observes that given a number of training steps, multi-epoch training on a randomly sampled dataset can achieve a better test performance than training with per-step fresh samples, for various algorithmic tasks [7]. In LLM post-training, a concurrent work [16] also observes

more epochs on fewer samples can lead to better performance under a fixed compute budget for math and coding tasks.

These are examples of what we refer to as *small-vs-large gaps*, where training on a smaller number of samples results in reduced training *compute* for a given model, where compute is defined as the total number of (possibly repeated) samples on which the model performs gradient updates (e.g., training steps × batch size) in order to reach a target performance.

This work aims to better understand such small-vs-large gaps. We begin by extending prior work, confirming that the small-vs-large gaps appear across a variety of settings (Figures 1 and 2), including different tasks, architectures, and optimizers, and under both mini-batch and full-batch updates. In contrast to prior studies, many of the settings we examine are not explained by existing theory (Section 4.1). These include comparisons of CSQ-SQ lower bounds [9, 3, 19], gradient variance reduction [17], and curriculum learning [24, 2] or learning under biased distributions [15, 8]. Notably, the small-vs-large gap exists even under full-batch gradient updates (Figure 2), implying that explanations based on stochastic gradient updates are not sufficient.

Instead, we show that the small-vs-large gap primarily results from favorable optimization biases due to **sampling biases** of the dataset. Intuitively, repeating the dataset reinforces the bias induced by sampling, which helps adjust the relative growth of different layers and in turn speeds up feature learning. This becomes more evident when the dataset is smaller due to a stronger sampling bias. We formalize this intuition in Section 4.2 and shows that training on smaller datasets can reduce the number of steps required for convergence (Theorem 1). This is further supported by the fact that the small-vs-large gap can be removed with proper selection of layerwise initialization or learning rates. We provide empirical evidence from various interventions in Section 5. Such sampling biases make the model more robust to learning rate and initialization choice, leading to a gap under standard parameterization.

In summary, our work characterizes the small-vs-large gap

[1]Department of Computer Science, Columbia University, New York, NY, USA [2]Department of Computer and Information Science, University of Pennsylvania, Philadelphia, PA, USA [3]Kempner Institute, Harvard University, Cambridge, MA, USA. Correspondence to: Jingwen Liu <jingwenliu@cs.columbia.edu>, Bingbin Liu <bliu@g.harvard.edu>.

*Proceedings of the 43$^{rd}$ International Conference on Machine Learning*, Seoul, South Korea. PMLR 306, 2026. Copyright 2026 by the author(s).

with the following contributions:

- We confirm that the small-vs-large gap exists across tasks, architectures, and optimizers. The gap is evident in both the number of *optimization steps* and the overall *compute complexity*, which depends on both the number of steps and the per-step cost, proportional to the batch size.

- We show that *sampling biases* induced by smaller datasets is a primary driver of the small-vs-large gap (Section 4): sampling biases modulate the relative magnitude of updates across layers, which in turn helps with feature learning. We identify regimes where existing theory fails to explain the gap (Section 4.1), and theoretically show that training on smaller datasets reduces the number of steps required for convergence in MLP (Theorem 1).

- We further support the theoretical explanation with a broad set of empirical evidence. First, training on a small dataset with *random labels* leads to a speedup comparable to that observed with real labels (Section 5.1), indicating that sampling bias is the main mechanism, as the gap persists without task-relevant signal. Moreover, *parameter-wise interventions* substantially reduce the small-vs-large gap (Section 5.2), including adjustments to initialization scales and parameter-wise learning rates across both MLP and Transformers. For Transformers, our findings additionally suggest that the widely used QK normalization has nuanced effects on optimization that merit further investigation.

Section 6 discusses implications and limitations. Together, our results suggest that training on a smaller dataset with increased repetitions is not merely a fallback under data scarcity, but a source of beneficial optimization inductive biases that can be leveraged more proactively, particularly for reasoning tasks.

## 2 Setup

**Tasks**  We consider synthetic tasks, which have tunable parameters and thus allow for explicit control over task complexity. We start with two classic feature learning which have been extensively studied in the literature.

- **Single-index model (SIM)**: the input is a Gaussian $x \sim \mathcal{N}(0, I_d)$ with label $y := \phi(\langle w^*, x \rangle)$, where $w^*$ is the ground truth feature vector, and $\phi : \mathbb{R} \to \mathbb{R}$ is an unknown link function. Our experiments take the link function to be a Hermite polynomial, denoted as $\text{He}_k$ for some order $k$.

- $(d, k)$-**sparse parity**: the input is a boolean $x \sim$ Unif($\{\pm 1\}^d$) with label $y := \prod_{i \in S} x_i$, where $S \subset [d]$ is an unknown support of size $k$.

Transformers consider two more algorithmic tasks:

- **In-context linear regression**: the input is a sequence $x_1, y_1, x_2, y_2, \ldots, x_k, y_k, x_q$ of length $2k+1$, where each sequence we independently sample a $w \sim \mathcal{N}(0, I_n)$, $x_i \sim \mathcal{N}(0, I_n)$, $y_i = w^\top x_i, \forall i \in [k]$ and the label is given by $y := w^\top x_q$.

- $(N, p)$-**modular addition**: the input are two numbers $x, z \sim$ Unif($[N]$), and the label is given by $y := (x + z) \mod p$ for some prime $p$. For Transformer experiments, $x, z$ are each represented by $\lceil \log_b N \rceil$ digits in base-$b$, and the output logits have size $p$.

**Data reuse strategies**  We consider both batch stochastic gradient descent (SGD) and (full-batch) gradient descent (GD) over datasets of different sizes. [1] For batch SGD, the batches are sampled uniformly over the distribution with replacement. We additionally consider **multi-phase training**, where the dataset sizes across phases can vary. In particular, for $T$-phase repeat, batches are sampled from a subset $\mathcal{S}_i$ at the $i_{th}$ stage for $i \in [T]$, where $\mathcal{S}_i \subset \mathcal{S}_j$ for $j > i$. [2]  An example is 2-phase training, where the first phase uses a subset randomly sampled from the population, and the second phase optimizes on the full population. This is similar to the two-set training proposed in [7], where each batch is a mix of samples from two sets: one small set which is repeated, and one large set consisting of online samples. General multi-phase training requires specifying the sizes and number of steps per phase. A heuristic is to make (1) the first few subsets relatively small so that the model can both quickly reach non-trivial train set performance and deviate non-trivially from initialization; and (2) the final subset $\mathcal{S}_T$ sufficiently large to ensure good generalization. We experiment with auto-scheduling following this heuristic as an ablation; details are provided in Appendix C.2.2 (Figure 19).

**Experimental setup**  Our primary focus is the performance under a given compute, which is measured by the batch size $\times$ number of optimization steps. We report *expected performance* under a fixed compute by taking the accuracy or loss averaged over random seeds. For tasks where the accuracy exhibits shape phase transitions, the average accuracy can also be interpreted as the *probability of success*.

---

[1] See Figure 18 for an ablation on the dataset size.

[2] We experimented with an alternative where each subsets are drawn independently without requiring to be a superset of the previous subsets. The results were similar, so we keep the subset requirement which has the additional benefit of smaller sample complexity.

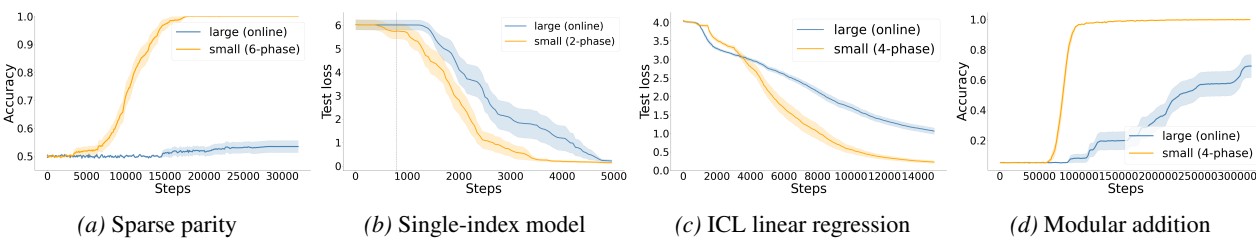

*Figure 1.* **Small-vs-large gap exists in various tasks**. Across various feature learning and algorithmic tasks (Section 2), training on a smaller dataset (yellow curves) leads to faster convergence than training on a larger dataset (blue curves). Results are based on 2-layer Transformers optimized with mini-batched AdamW. An "$n$-phase" schedule denotes that the training set size is progressively increased over $n$ phases (Section 2).

We train with both MLPs and Transformers. The MLP has ReLU activation and no residual connections. The Transformer has an optional QK normalization. Models are of depth-2 unless specified otherwise. All weights are initialized with Pytorch defaults; for example, $W_{ij} \sim$ Unif$[-1/\sqrt{d_{\text{in}}}, 1/\sqrt{d_{\text{in}}}]$. We use the SGD optimizer for MLPs and AdamW for Transformers unless specified, and sweep over the learning rate for each setup. Details are provided in Appendix C.

## 3 Small-vs-large gap: less data can lead to faster learning

We first present empirical evidence that less data leads to accelerated learning across tasks and setups.

**Mini-batch updates** We start with training using mini-batch updates, which is the common training strategy in practice and where prior work has also reported the small-vs-large gap [7]. As shown in Figure 1, smaller datasets lead to faster convergence for all tasks. For SIM, in-context learning regression and modular addition, multi-phase is used to balance accelerated optimization and good generalization.

However, mini-batch updates introduce a confounding factor of the *number of repetitions*: when trained for the same number of steps, each sample in a smaller dataset is reused more frequently over the course of training. It is therefore unclear whether this increased repetition is the primary source of the observed speedup. Our subsequent gradient descent results show that this is not the case.

**Gradient descent (full-batch updates)** We sample datasets of varying sizes from the population and run (full-batch) gradient descent on each dataset. As shown in Figure 2, smaller datasets have better performance at each time step throughout training. Moreover, the total saving in compute is much more significant than the reduction in steps, since smaller datasets also incur lower per-step computational cost. For instance, for $(20, 6)$-sparse parity, using $N = 2^{14}$ converges in 1500 steps whereas using train-

ing on the full population (i.e. $N = 2^{20}$) requires more than 2000 steps, leading to a 100x speedup in compute.

**Remark** (Population gradients for sparse parity). Sparse parity has a special structure that the population gradient of the first layer weight reveals information about the support [6], hence training with the full population can in theory leverage this information and converge quickly. However, doing so requires large learning rate (on the order of $d^k$) which is infeasible due to numerical limitations and dynamics for the second layer. Our experiments confirm this and we did not see strict speedup from increasing the learning rate.

## 4 Unpacking the efficiency gain from smaller datasets

This section focuses on sparse parity as a sandbox to understand the small-vs-large gap. We first explain why existing work is not sufficient, and then provide our explanation that hinges on dataset sampling biases.

### 4.1 Prior theories are insufficient

**SQ-CSQ difference.** One mechanism of acceleration identified by prior work is that taking multiple gradient updates on the same data effectively transform (batch) SGD from a correlational statistical query (CSQ) algorithm to a statistical query (SQ) algorithm [9, 3, 19], with the latter being a more powerful class of algorithms. Prior work uses this to explain the speedup for batch SGD on single-index model (SIM), for which there is a known gap between CSQ and SQ lower bounds. While insightful, the CSQ-SQ gap cannot fully explain the observed speedup. It cannot explain why there is a speedup for SIM even when using (full-batch) GD, where smaller and larger datasets are both repeated and for the same amount of times. Moreover, it is not applicable to the wide range of tasks where the SQ and CSQ lower bound coincide, which include all discrete problems such as sparse parity and mod addition.

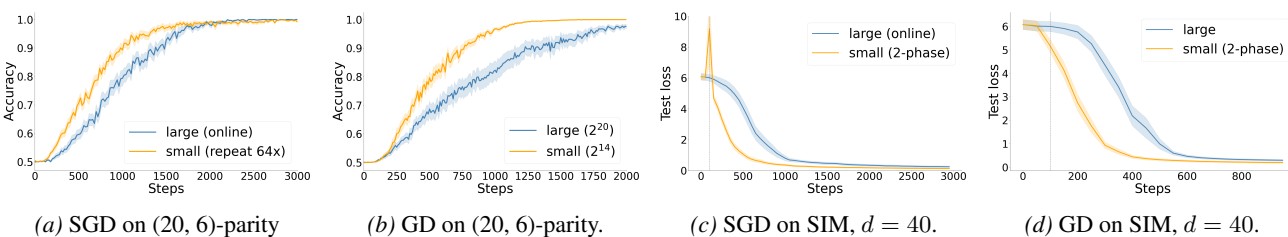

*(a)* SGD on (20, 6)-parity     *(b)* GD on (20, 6)-parity.     *(c)* SGD on SIM, $d = 40$.     *(d)* GD on SIM, $d = 40$.

*Figure 2.* **Small-vs-large gap exists for both mini-batch and full-batch training**. Results are based on SIM and parity with 2-layer MLPs, optimized with both mini-batch (SGD) and full-batch (GD) updates. The small-vs-large gap with GD is a notable example that prior theory fails to capture (Section 4.1).

**Gradient variance reduction.** It is known that reducing gradient variances help speed up convergence [14], and more recently [17] has reported that reducing gradient variances across batches can accelerate training in sparse parity and in-context linear regression tasks. However, the variance reduction point of view cannot explain the speedup with full-batch updates, where each step uses the full dataset and hence has no sampling-induced variances.

**Biased (input) distribution.** Specific to sparse parity and SIM, prior theory indicates another explanation from the benefit of biased distributions. The focus was on the biases in the input distribution, which can either be explicitly constructed sparsity [24, 2], or randomly perturbed distributions [15, 8]. Our setup is closer to the latter due to the deviation of the empirical mean from the population mean. However, a critical difference is that the signal strength (in terms of first-order Fourier coefficients) in [8] is exponential in the sparse $k$, whereas the sampling bias in our analysis depends on the dataset bias only and is independent of the sparsity. In particular, for a random subset of size $N$, the signal strength in [8] is $O(N^{-k/2})$, which is much smaller than the $O(N^{-1/2})$ sampling bias as detailed in Section 4.2.

We additionally provide two pieces of empirical evidence that the input distribution biases are not the primary driver of the gap: the speedup persists even when the dataset input bias is removed, and that online training with biased input distribution does not lead to the same amount of speedup.

Small datasets without biases still lead to speedup. We show that the small-vs-large gap exists when we remove input biases in the small datasets. For parity, we ensure the training set satisfies $\hat{\mathbb{E}}[x] = 0$, and optionally further requiring $\hat{\mathbb{E}}[y] = 0$ and $\hat{\mathbb{E}}[x|y = -1] = \hat{\mathbb{E}}[x|y = 1] = 0$. The small-vs-large gap persists as shown in Figure 15(a). Consistent results are observed on Transformer for both parity and SIM (Figure 16). For SIM, we whiten the inputs to match both 1st and 2nd order statistics, i.e. transform the dataset with $\tilde{x} = \hat{\Sigma}^{-1/2}(x - \hat{\mu})$, where $\hat{\mu}$ and $\hat{\Sigma}$ are the empirical mean and covariance.

Biased online training has minor effects. On the other hand, we train with freshly sampled online data, whose per-coordinate Bernoulli parameters are set to the biases of a finite offline dataset. Specifically, for $d = 20$, $k = 6$, we take the biases from $2^i$ samples for $i \in \{4, 6, 8, 10, 12\}$. [8] indicates that using biasing the distributions will improve training speed, which we also confirm in Figure 15(b). However, unless the samples size is exceedingly small (e.g., fewer than $2^5 = 32$ samples), the speedup is much more moderate compared to training on a small subset, and the small-vs-large gap persists.

### 4.2 Our explanation: dataset sampling bias accelerates learning by adjusting the relative norm growth

We claim that a primary source of smaller datasets' speedup is their sampling biases, which adjust the norms across layers and effectively change the per-layer learning rates. Intuitively, for 2-layer MLPs learning parity, feature learning occurs in the first (i.e., input) layer. Growing the second layer speeds up the feature learning in the first layer, and smaller datasets have stronger biases that grow the second layer faster. The rest of this section formalizes this intuition. We provide empirical evidence in Section 5.

For the theoretical analysis, we consider learning the 2-sparse parity task with a 2-layer network with quadratic activation, i.e. $f(x) = a\sigma(w^\top x) - 1$, with $\sigma(z) := \frac{1}{2}z^2$. The model is optimized using the correlation loss, i.e. $L(f) = \mathbb{E}_{x,y}[\ell(y, y')]$, where $\ell(y, y') = -yy'$. Let $\eta$ denote the learning rate. We consider projected updates

$$a^{(t+1)} = \max\{-1, \min\{1, a^{(t)} - \eta\nabla_{a^{(t)}}L\}\}, \quad (1)$$

$$w^{(t+1)} = \frac{w^{(t)} - \eta\nabla_{w^{(t)}}L}{\left\|w^{(t)} - \eta\nabla_{w^{(t)}}L\right\|_2}. \quad (2)$$

Following standard practices, we initialize with $w^{(0)} \sim \text{Unif}(\mathbb{S}^{d-1})$, and $a^{(0)} \sim \mathcal{N}(0, 1/m)$ where $m$ can be considered as a model width parameter. We focus on the gradient descent (GD) process; the SGD process is a noisy version of GD and can be analyzed using techniques from [4, 1]. Since the correlation loss does not introduce

interaction across neurons, the analysis below can be considered as focusing on one neuron of a width-$m$ network. Without loss of generality, we have the support on the first 2 coordinates and the minimizer $w^\star$ has $|w_1^\star| = |w_2^\star| = \frac{1}{\sqrt{2}}$, and $w_i = 0$ otherwise.

We show that smaller $N$ leads to faster convergence:

**Theorem 1** (2-phase training from standard initialization)**.** *Consider a 2-phase training with $m > d$ and learning rate $\eta = \Theta(1)$. The first phase updates with Equation* (1) *using a randomly sampled dataset of size $d \leq N \leq d^2$, until $|a| \geq a_\star$ for some $0 < a_\star \lesssim \frac{1}{(Nd)^{1/4}(\log d/\delta)^{1/2}}$; the second phase updates with Equation* (1) *using the full population gradient, until reaching a $\hat{w}$ such that $\|\hat{w} - w^\star\|_2 \lesssim \sqrt{\varepsilon}$. Let $T_1, T_2$ denote the numbers of steps required in each phase respectively. Let $p_{all} \in (0,1)$ be a universal constant where $p_{all} = \Theta(1)$.* [3] *Then, with probability at least $p_{all} - \delta$ over the random initialization and the phase-1 samples,*

$$T_1 \lesssim \frac{a^* \sqrt{N}}{\eta}, \quad T_2 \lesssim \frac{2}{\eta a^*} \log\left(\frac{d}{\varepsilon}\right). \qquad (3)$$

*With the optimal choice of $a_\star$, the total number of steps is $O\left((Nd)^{1/4} \log\left(\frac{d}{\varepsilon}\right) \log^{1/2}\left(\frac{d}{\delta}\right)\right)$.*

Theorem 1 implies that a small $N$ leads to a direct saving in the number of steps $T$. One could alternatively skip Phase 1 and train directly on the full population. This will require $O(m^{1/2} \log(d/\epsilon))$ steps (Lemma 6), which is worse than the 2-phase convergence when $m \gg d^2$.

*Proof sketch.* The gradient magnitude of $w$ depends on the magnitude of $a$, and since $a$ is initialized to be very small $(O(1/\sqrt{m}))$, this slows down the learning of $w$. However, the sampling bias of small datasets can quickly grow the magnitude of $a$ at the initial stage of training. The gradient of $a$ is given by $q^{(t)} := (w^{(t)})^\top \widehat{M} w^{(t)}$, where $\widehat{M} := \widehat{\mathbb{E}}[yxx^\top] := \frac{1}{N}\sum_{s=1}^{N} y^{(s)} x^{(s)} x^{(s)\top}$. Due to the anti-concentration of $\widehat{M}$, $q^{(t)}$ is on the order of $N^{-1/2}$, whereas the population quantity is on the order of $1/d$. Therefore, in the first phase, $a$ grows at a linear rate of $N^{-1/2}$, hence the number of steps for $a$ to reach $a_\star$ is proportional to $N^{1/2}$. Once $a$ reaches $a_\star$, we switch to the second phase which uses population updates. The analysis is on the power iteration on the true moment $M$, whose contraction rate depends on $\eta a_\star$.

In fact, the first-phase analysis primarily relies on the anti-concentration of the empirical moment matrix $\widehat{M}$, which is largely independent of the true label signal. This suggests that the early-stage acceleration should still persist even when the labels are random. We further study this in Section 5.1. Furthermore, the convergence rate of the fea-

---

ture direction $w$ depends on the strength of its gradient signal relative to its scale. The second-phase analysis reveals that the strength of the gradient is governed by its learning rate and the initial magnitude of $a$. This directly motivates empirical interventions explored in Section 5.2, where we manipulate per-layer learning rates and layer-wise initialization scales.

Notably, the $N^{-1/2}$ bias in Phase 1 contrasts with results in [8], where a per-coordinate bias $\eta$ induces a Fourier coefficient on the order of $\eta^k$, which is non-negligible only when $k$ is bounded. In contrast, our analysis is independent of $k$, suggesting that the small-vs-large gap persists even in dense regimes where $k$ can be as large as $d$, as confirmed empirically (Figure 12). Moreover, the notion of "bias" differ between the two settings. [8] assumes $\eta = \Omega(1)$, which is substantially larger than the $O(N^{-1/2})$ sampling bias unless $N$ is unreasonably small, as discussed in the biased online training part.

## 5 Empirical evidence for relative norm growth

This section provides empirical evidence supporting the claim in Section 4 that less data leads to faster learning by affecting the relative growth rates of the two layers. Figure 10 provides direct observational evidence: during the initial phase of training, the weight norm ratio $\frac{\|a\|_2}{\|W\|_F}$ increases more rapidly when the dataset is smaller, for both parity and SIM.

We further consider *interventions* on the training process to provide stronger evidence, in terms of 1) *data*, where the speedup from small-set exists even when the labels are random; 2) *weight norms*, by changing initialization scale or the adoption of normalization layers; and 3) layer-wise *learning rate* controls. All results support our hypothesis, as detailed below.

### 5.1 Small-vs-large gap exists when training first on *random* labels

One implication of Theorem 1 is that small dataset biases help grow the second layer faster, which in turn amplifies the update in the input layer which is responsible for feature learning. This suggests that any methods that help grow the second layer should achieve similar acceleration, even when the growth results from signals unrelated to the target function.

Training with *random labels* is one example that provides such a speedup. Specifically, consider a modified 2-phase training procedure similar to Theorem 1, where we alter the first phase to train with random labels, i.e., $y$ is sampled

i.i.d. from Uniform$\{-1, +1\}$. We show that the gradient of the second layer $(a)$ is still of order $\Theta(N^{-1/2})$ around initialization.

**Corollary 2** (2-phase training, with Phase 1 on random labels). *Consider the setting in Theorem 1 where the first phase uses a randomly sampled dataset of size $d \leq N \leq d^2$, with labels uniformly sampled from $\{-1, +1\}$. Similar to Theorem 1, let $p_{\text{all}} \in (0, 1)$ be a universal constant where $p_{\text{all}} = \Theta(1)$. Then, with probability at least $p_{\text{all}} - \delta$ over the random initialization and the phase-1 samples, the total number of steps required to reach a $\hat{w}$ such that $\|\hat{w} - w^\star\|_2 \leq \varepsilon$ is*

$$T = O\left(\frac{\sqrt{N}}{\eta\sqrt{d}} + \frac{\sqrt{d}}{\eta}\log\left(\frac{d}{\varepsilon}\right)\right).$$

We verify the speedup empirically by training a MLP first on a small set of *random* labels, and then switching to the large-set training. Our hypothesis will be supported if the first phase on random labels leads to accelerated learning of the actual task. We experiment with MLP on parity and SIM: for parity, the random labels are obtained by sampling $y$ uniformly from $\{-1, +1\}$; for SIM, we sample a random feature vector $w_{\text{random}} \sim \mathcal{N}(0, I)$ and use $w_{\text{random}}$ along with the true link function to label the small dataset. As shown in Figure 3, both the accuracy and the weight norm growth (measured by $\|a\|_2/\|W\|_F$) are sensitive only to the dataset size, but not the label choice (i.e., true or random labels). This also agrees with our theory (i.e., phase 1 analysis) that the benefit of sampling bias exists even with random labels. We observe similar results for Transformer learning mod addition, where an initial phase of random label training speeds up the subsequent actual learning (Figure 20)

### 5.2 Small-vs-large gap diminishes with parameter-wise interventions

This section investigates direct interventions on layer-wise scalings. We find that, consistent with the analysis in Section 4.2, these interventions can significantly reduce and sometimes eliminate the small-vs-large performance gap. However, smaller sets exhibit favorable optimization biases that make training more robust to hyperparameter choices. Results are reported on MLPs trained with the SGD optimizer and Transformers trained with AdamW. [4]

#### 5.2.1 MLP: LAYER-WISE INTERVENTIONS

One takeaway from Section 4.2 is that the small-vs-large gap comes from the fact that smaller-set training can bet-

---

[4]AdamW affects MLP and Transformer training differently, which we discuss in more details in Appendix C.2.2.

ter adjust the relative norms across layers. Thus, interventions that directly adjust the relative layer norms should be able to reduce the gap. Such relative norm change can be achieved in two ways: 1) at initialization, by directly adjusting the standard deviations from which the weights are sampled from, and 2) throughout training, by explicitly supplying layer-specific learning rates.

For **initialization scales**, we multiply the initial weights by layer-specific constants. Figure 4 shows the final iterate test accuracy for a large sweep over scales for a 2-hidden-layer MLP, where we consider 2 scaling parameters, one for the first layer weights and one for the other layers. The results show that there exist scaling constants that completely close the small-vs-large gap between size-$2^{14}$ random subsets and the full population ($2^{20}$). In particular, the larger dataset performs worse at the default init (marked by the red star), but shrinking the first layer scale and growing the other layers improves its performance and eventually makes the gap vanish. However, identifying the right constants requires searching through a large set of combinations, whereas small-set training can adjust the scaling automatically and is much more robust to initialization scale.

For MLP with ReLU activations, growing one layer leads to larger updates on another. Therefore, shrinking the input layer and scaling up the output layer can be effectively considered as using **layer-wise learning rates** which are larger for the input layer and smaller for the other. The empirical evidence agrees with this hypothesis. Let $\eta_1, \eta_2$ denote the learning rates for the first and second layer, respectively. Figure 5 shows that the optimal choice of $(\eta_1, \eta_2)$, where $\eta_1 \gg \eta_2$, significantly reduces the small-vs-large gap between the full population ($N = 2^{20}$) versus a random subset ($N = 2^{14}$).

***What is the optimal scaling?*** The above results show that the small-vs-large gap can be bridged when using proper layerwise initialization scaling or learning rates. It is then desirable to identify such a scheme without extensive hyperparameter search. A natural candidate is the $\mu P$ parameterization [29, 30], which is designed to maximize feature learning. Empirically, we find $\mu P$ to be a strong starting point, bridging the gap in 2-layer MLPs for parity, yet the small-vs-large gap still exists for SIM (Figure 13). Following the argument in Theorem 1, we additionally consider a one-dimensional search of a single parameter $\alpha$, which adjusts initialization scaling by dividing the first layer's standard deviation by $\alpha$ and multiplying the second layer's by $\alpha$. As shown in Figure 13, such $\alpha$ scaling suffices to bridge the small-vs-large gap in both parity and SIM, and the optimal $\alpha$ remains *constant* across model widths (Figure 23).

One could potentially consider a more thorough search on

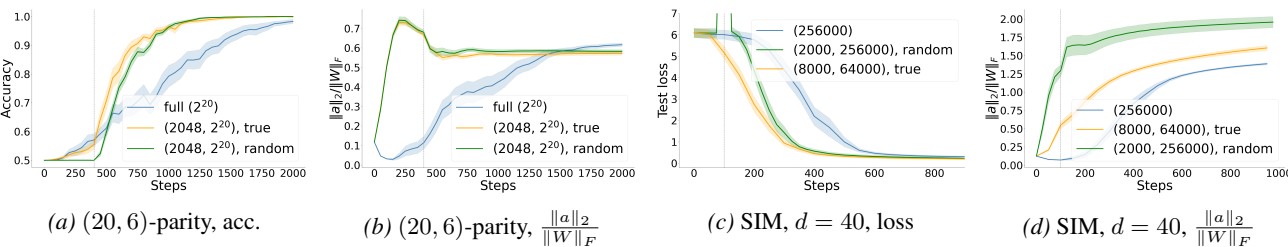

*(a) (20, 6)-parity, acc.*     *(b) (20, 6)-parity, $\frac{\|a\|_2}{\|W\|_F}$*     *(c) SIM, $d = 40$, loss*     *(d) SIM, $d = 40$, $\frac{\|a\|_2}{\|W\|_F}$*

*Figure 3.* **Training on small datasets with *random labels* leads to faster learning**. For GD on both parity and SIM, the initial random-training leads to significant speedup and faster growth of $\|a\|_2/\|W\|_F$. The blue/yellow curves correspond to large/small sets. The green curves correspond to training first on a small set of *random* labels and then switching to large sets with true labels.

the design choices of layerwise initialization or learning rates scheme, for which there is a vast existing literature as discussed in Appendix A. Our results on the small-vs-large gap provide an orthogonal angle, suggesting that *data use strategy* can offer helpful inductive biases that make the model robust to the scale or learning rates.

### 5.2.2 Transformer: scaling of $W_q, W_k$

The small-vs-large gap is observed in Transformers for both mini-batch updates (Figure 1) and full-batch updates (Figure 14). Similar to MLP, we find that the gap can be attributed to favorable optimization biases induced by small-set training, and proper interventions can reduce the gap. We focus on interventions on $W_q, W_k$ matrices which directly affect attention, and provide ablation studies on other parameters in Appendix C.2.2.

The small-vs-large gap is reduced when increasing the *learning rate* for $W_q, W_k$ or increasing the *initialization scale* of $W_q, W_k$ for parity (Figure 6). Specifically, for the learning rate intervention, we tune a separate learning rate for $W_q, W_k$ while keeping the learning rate for other parameters fixed at the optimal value for the global learning rate. In Figure 6a, the optimal QK learning rate is 3.6x that of the optimal global learning rate for large-set training (i.e. using online mini-batches), 2.5x for the small-set training with 100 epochs, and 1.25x for the small-set 6-phase training. For QK initialization, large-set training benefits greatly from increased initialization scale; the optimal scale is 8x the default initialization, which sharpens the attention logits by a factor of 64 when considering both $W_q, W_k$. In contrast, small-set training gets much more moderate improvements from tuning the initialization scale: 100-epoch training gets the best speedup when scaling the default by 2x, while 6-phase training sees no benefit from scaling changes. Similar phenomena are observed in SIM and ICL as well (Figure 7), where tuning the QK initialization narrows (but not necessarily closes) the small-vs-large gap.

**Connection to and implications of QK normalization**
The attention logits scaling adjustment mentioned above

is reminiscent of QK normalization, which normalizes the key and query vectors to a sphere [12, 10, 32]. For example, the commonly adopted RMSNorm imposes a scaling of $\asymp \sqrt{d}$ compared to default non-normalized version. QK normalization is now common practice for preventing training instabilities, as it constrains the attention logit magnitudes [10, 32, 25].

Our results suggest that its implications on optimization may be more nuanced. On the plus side, QK normalization can significantly speed up optimization for online training for sparse parity and SIM, removing the initial saddle-like behavior and almost entirely closing the small-vs-large gap (Figure 6c, Figure 7b). However, such acceleration is not universal, and we see QK normalization hurting both online and subset training for ICL (Figure 7d) and mod addition (Figure 27b). Moreover, QK normalization can exacerbate overfitting under data repetition, which is observed in both mini-batch (Figure 6c, 6-phase) and full-batch training (Figure 27a); a train-test comparison is provided in Figure 28. Understanding the exact mechanism is left as future work.

## 6 Discussions

This work studies the small-vs-large gap, a phenomenon that under a fixed compute budget, repeating on smaller dataset outperforms training on larger dataset. We identify the sampling bias from smaller datasets as a key mechanism underlying this effect, which helps adjust the relative norms across layers. We formalize this mechanism theoretically and substantiate it with empirical results. Notably, across tasks and for both MLP and Transformers, we find that the small-vs-large gap persists even when training on randomly labeled data, and that proper choices of initialization scale and layer-wise learning rates can substantially reduce or even close the gap.

Beyond the small-vs-large gap itself, our results suggest that dataset size and repetition can be leveraged as favorable optimization biases. In particular, repeated training on smaller subsets can act as an implicit layerwise precon-

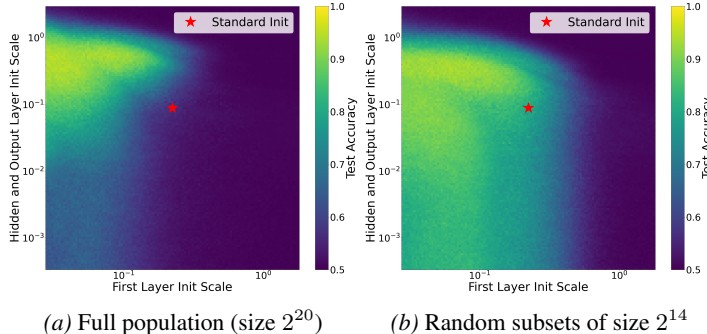

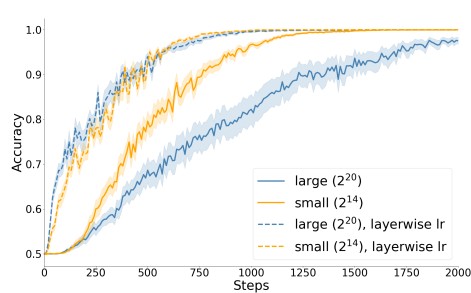

*(a)* Full population (size $2^{20}$)   *(b)* Random subsets of size $2^{14}$

*Figure 4.* **Proper initialization removes the small-vs-large gap**, though smaller-set training is more robust to the initialization scale. Results are shown for $(20, 6)$-parity with MLP. The heatmaps show the accuracies (averaged over 256 seeds) using per-setup best learning rate.

*Figure 5.* **Layer-wise learning rate removes the small-vs-large gap.** The optimal learning rate for $w$ is larger than that of $a$. Results are shown for MLP on $(20, 6)$-parity, trained with GD.

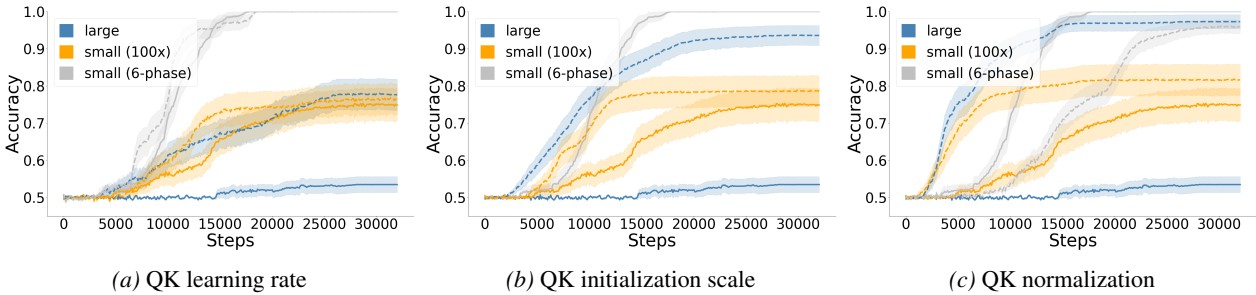

*(a)* QK learning rate   *(b)* QK initialization scale   *(c)* QK normalization

*Figure 6.* **Small-vs-large gap in Transformers can be reduced with interventions on $W_Q, W_K$.** Results are shown for $(20, 6)$-parity with two-layer Transformers; similar results are also observed for SIM and ICL (Figure 7). The small-vs-large gap is reduced by (a) tuning learning rate on $W_Q, W_K$ separately than the rest of the parameters; (b) using the optimal initialization scaling of $W_Q, W_K$; (c) using QK-layernorm. Solid lines are the default setup where we observe clear small-vs-large gaps, and dashed lines are interventions that reduce or even revert the small-vs-large gap. Interventions are not helpful especially for 6-phase training: for (b), the optimal initialization scaling is the default one, hence we omit the corresponding dashed line; for (c), QK layernorm slows down 6-phase training.

ditioner that steers models into a more favorable feature-learning regime, partially substituting for carefully tuned initialization, normalization, or layerwise learning rates. This perspective motivates training pipelines that explicitly separate an early "optimization-shaping" phase from a later "coverage/generalization" phase, and suggests that compute-optimal training may require jointly considering data, optimizer, and parameterization.

Below we discuss other implications of our findings.

**How to enlarge the small-vs-large gap.** We investigate how the small-vs-large gap varies as we scale along different axes. The following factors are found to widen the gap:

- *Increasing model depth.* The small-vs-large gap is due to the relative norm growth across layers, which suggests that the gap should be more pronounced when the model depth increases as the scaling effect will percolate exponentially in depth. Our empirical findings confirm this, where the small-vs-large gap widens for both MLP (Figure 9a) and Transformers (Figure 26).

- *Reducing model width.* We find that the small-vs-large gap is the widest at a small model width (e.g., width 64), as shown in Figure 9b for parity and Figure 24 for SIM. We hypothesize this is because models with standard initialization approach the kernel regime as the width increases, suggesting that the small-vs-large gap is specific to feature learning.

- *Increasing task complexity.* More difficult tasks lead to a wider small-vs-large gap, across tasks and architectures. Figure 8a shows SIM results on MLP trained with full-batches, where the gap is wider on $d = 80$ than $d = 40$. Figure 8b shows parity results on Transformers trained with mini-batch updates, where $(20, 6)$-parity sees a bigger gap than $(10, 6)$-parity.

- *Smoother transition between training phases.* Compared to the 2-phase analysis in Section 4.2, growing the repeating subset size more gradually helps improve training and hence increases the small-vs-large gap. In Figure 6, training transformer using 6-phase training greatly enlarges the gap comparing to 1-phase training in parity.

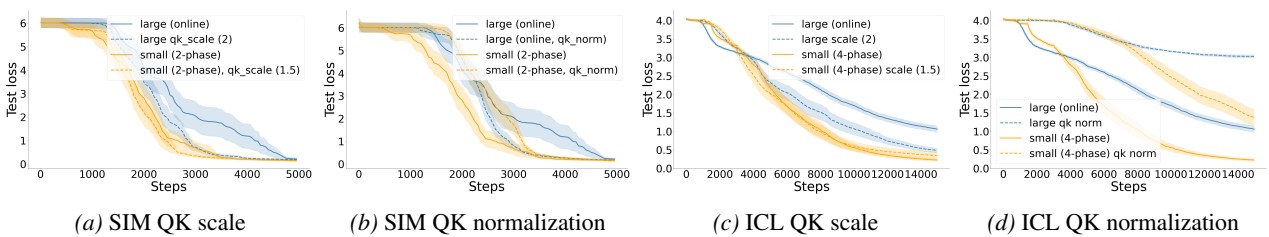

*(a)* SIM QK scale     *(b)* SIM QK normalization     *(c)* ICL QK scale     *(d)* ICL QK normalization

*Figure 7.* **Small-vs-large gap in Transformers can be reduced with interventions on** $W_Q, W_K$, for (a) SIM and (b) in-context learning regression trained using two-layer Transformers. Solid lines are the default setup where we observe clear small-vs-large gaps, and dashed lines are interventions that reduce or even revert the small-vs-large gap.

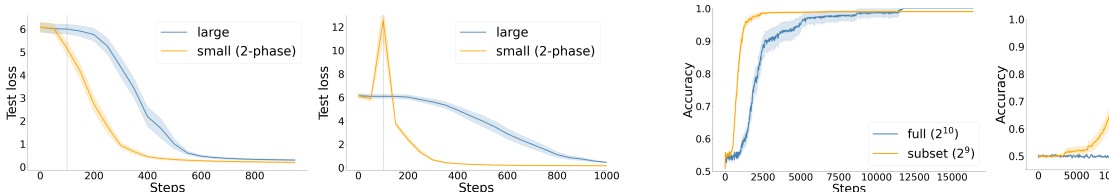

*(a)* MLP on SIM with dimension $d = 40$ (left) and $d = 80$ (right), trained with GD.

*(b)* Transformer on parity with dimension $d = 10$ (left) and $d = 20$ (right), trained with batched AdamW.

*Figure 8.* **Increasing the task complexity** increases the small-vs-large gap.

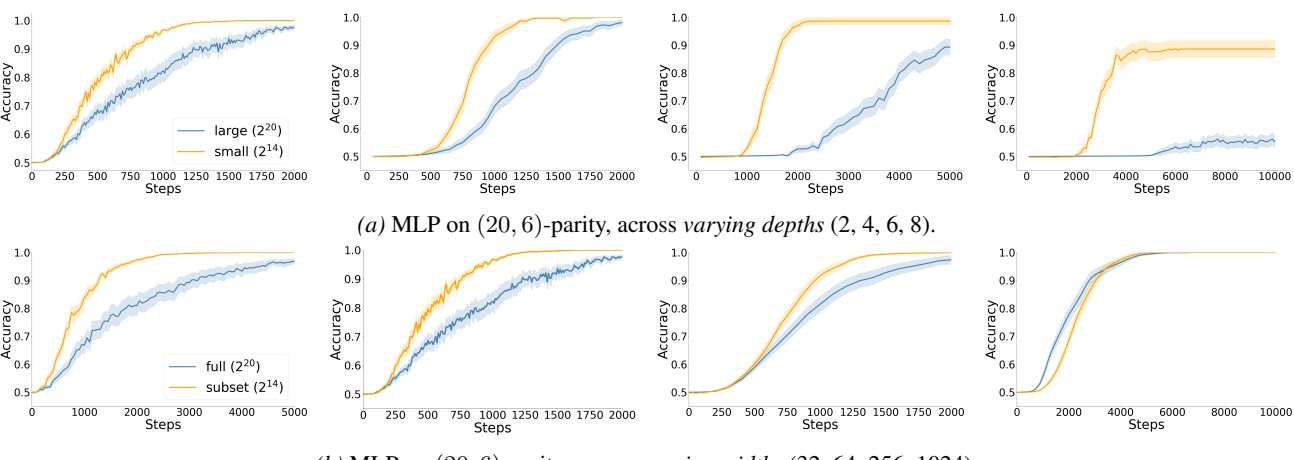

*(a)* MLP on $(20, 6)$-parity, across *varying depths* $(2, 4, 6, 8)$.

*(b)* MLP on $(20, 6)$-parity, across *varying widths* $(32, 64, 256, 1024)$.

*Figure 9.* **Model sizes affect the small-vs-large gap**. Increasing model depth widens the gap (*top row*), whereas increasing the model width reduces the gap (*bottom row*). Results are shown on sparse parity learned with MLP using full-batch updates; similar results are also observed on SIM with MLP (Figure 24) and parity with Transformers (Figure 26).

**When is data repetition (not) helpful?** Even though the small-vs-large gap has been observed across various choice of tasks, architectures, and optimizers, we do not believe it to exist universally. A classic example is linear regression, which does not have a small-vs-large gap. While prior work has shown that multi-epoch training improves the statistical complexity for linear regression [22, 20], no result has suggested an improvement in terms of optimization steps or the compute cost. We hypothesize that the speedup from data repetition is exclusive for *non-convex* optimization, and a *computational-statistical gap* is likely required. Connecting to practice, especially the era of large-language model training, we do not expect this phenomena to be directly observable in many real-world language corpora, due to both (near) duplicates widely present in the web datasets, and the lack of clear structure in free-form texts. However, we hypothesize that the small-vs-large gap may be of interest for more *structured tasks* such as formal reasoning. As empirical evidence, the concurrent work [16] observes the small-vs-large gap in LLM post-training for math and coding tasks.

## Acknowledgment

We thank Alex Damian, Aditi Raghunathan, Andrej Risteski, Daniel Hsu, Eric Wong, and Samuel Deng for helpful discussions and feedback. JL is supported by NSF award DMS-2502259 and ONR N00014-22-1-2713. SG was supported in part by an AI2050 Early Career Fellowship from Schmidt Sciences. BL was supported by the Kempner Fellowship from the Kempner Institute at Harvard. This work was enabled in part by a gift from the Chan Zuckerberg Initiative Foundation to establish the Kempner Institute for the Study of Natural and Artificial Intelligence. Part of this work was done when the authors were participating in the Special Year on Large Language Models and Transformers and the Program on Modern Paradigms in Generalization at the Simons Institute for the Theory of Computing at UC Berkeley.

## Impact Statement

This paper presents work whose goal is to advance the field of machine learning. There are many potential societal consequences of our work, none of which we feel must be specifically highlighted here.

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

## A   Related work

It is widely believed that in deep learning, more is better, as captured by the study of scaling laws. However, different resources may not need to be scaled together. For instance, *data repetition*, which keeps the sample size fixed while scaling up compute, can achieve similar performance to compute-matched online training when the amount of repetition is moderate [27, 23, 21, 20, 28]. We are interested in the more extreme phenomenon termed the *small-vs-large gap*, where *reducing* the sample size when holding compute constant can help improve performance. The small-vs-large gap has been observed in recent work on algorithmic tasks [7], in-context learning [33], and language model finetuning on reasoning tasks [16]. Prior work has shown this for learning single-index models, where taking more than one gradient steps on the same set of samples can reduce the total number of gradient steps [9, 3, 19]. The intuition is that while online SGD lies in the class of correlational statistical query (CSQ) algorithms, SGD with sample repeats belongs the more general class of statistical query (SQ) algorithms. In contrast, we find that the compute savings from using less data hold even in regimes where the CSQ-SQ distinction does not apply, including training with full-batch gradient descent, and tasks with discrete domains. Close to our quadratic setting in Section 4.2, a concurrent work by [18] shows the statistical advantage of full-batch gradient descent over SGD where mini-batches are sampled fresh from the population. A key difference is that the model in [18] only has a single layer (i.e., $f(x) = \sigma(w^\top x)$), hence the effect of relative weight norm does not apply.

Previous work has also studied how multi-pass SGD can improve the sample complexity over single-pass SGD in various settings, including linear regression [22, 20], general stochastic convex optimization [23], and non-convex problems under the PL condition [27]. A crucial difference from our work is that these results focus on saving *samples* but not the *compute*: they show that the population error achieved by $T$ online SGD steps, where each step is taken on an iid sample from the population, can be achieved by $T$ steps of multi-pass SGD, where each step is taken on an iid sample drawn from an empirical distribution of size smaller than $T$. In contrast, we will show that it is possible to achieve the same error as $T$ online SGD steps using *fewer than $T$ steps* of multi-pass SGD.

We will show that the key mechanism behind such speedup comes from the strong sampling biases from smaller datasets, which effectively adjusts the relative update speeds across the layers, leading to faster learning. Such adjustments relate to the idea of balancing contributions from different layers, which has been studied extensively in the optimization and feature learning [29, 5, 30, 31, 11].

## B   Proof details

Motivated by the empirical evidence in Sections 3 to 5, we analyze a minimal setting, a single quadratic neuron on 2-sparse parity under a two-phase schedule: phase 1 runs GD on a fixed dataset of size $N$, and phase 2 switches to the full population.

### B.1   Setup

We assume the input $x \in \{\pm 1\}^d$ is sampled uniformly from the hypercube and the label $y = x_1 x_2 \in \{\pm 1\}$ is a 2-sparse parity. We study the quadratic neuron

$$f(x) = \frac{1}{2}a(w^\top x)^2,$$

trained with correlation loss $\ell(y, \hat{y}) = -y\hat{y}$. Let $w^\star$ denote a global minimizer, where $|w_1^\star| = |w_2^\star| = \frac{1}{\sqrt{2}}$, and $w_i = 0$ otherwise.

**Projection.**   To ensure that the weights are bounded at the solution, we use the following projections:

- **Output weight clipping:** after each update we set $a \leftarrow \text{clip}_{[-1,1]}(a)$.

- **Input weight renormalization:** after each update we set $w \leftarrow w/\|w\|_2$.

We consider 2-phase training, where Phase 1 uses a randomly sampled dataset of size $N$, and Phase 2 uses the full population.

**Phase 1 (fixed batch of size $N$).**  Fix a dataset $\{(x^{(s)}, y^{(s)})\}_{s=1}^N$ and define the empirical moment matrix

$$\widehat{M} := \widehat{\mathbb{E}}[yxx^\top] := \frac{1}{N} \sum_{s=1}^N y^{(s)} x^{(s)} x^{(s)\top}.$$

One step of (projected) gradient descent on this fixed batch takes the form

$$a^{(t+1)} = \mathrm{clip}_{[-1,1]}\Big(a^{(t)} + \frac{\eta}{2}(w^{(t)})^\top \widehat{M} w^{(t)}\Big), \tag{B.4}$$

$$w^{(t+1)} = \frac{w^{(t)} + \eta a^{(t)} \widehat{M} w^{(t)}}{\left\| w^{(t)} + \eta a^{(t)} \widehat{M} w^{(t)} \right\|_2}. \tag{B.5}$$

Let's define

$$q^{(t)} := (w^{(t)})^\top \widehat{M} w^{(t)}$$

which will be useful in the analysis.

**Phase 2 (population).**  After phase 1, we switch to population gradients, replacing $\widehat{M}$ by the population matrix

$$M := \mathbb{E}[yxx^\top] = e_1 e_2^\top + e_2 e_1^\top.$$

in the above updates.

We restate Theorem 1 below, which shows that using a smaller $N$ in Phase 1 improves convergence.

**Theorem** (2-phase training from standard initialization; Theorem 1 restated.).  *Consider a 2-phase training with $m > d \geq 3$ [5] and learning rate $\eta \leq \frac{1}{2}$. The first phase uses a randomly sampled dataset of size $d \leq N \leq d^2$, until $|a| \geq a_\star$ for some $a_\star \in (0,1)$ where $a_\star \lesssim \frac{1}{(Nd)^{1/4}\sqrt{\log(d/\delta)}}$; the second phase uses the full population gradient, until reaching a $\hat{w}$ such that $\|\hat{w} - w^\star\|_2 \lesssim \sqrt{\varepsilon}$. Let $T_1, T_2$ denote the numbers of steps required in each phase respectively. Let $p_{\mathrm{all}} \in (0,1)$ be a universal constant where $p_{\mathrm{all}} = \Theta(1)$. [6] Then, with probability at least $p_{\mathrm{all}} - \delta$ over the random initialization and the phase-1 samples,*

$$T_1 \lesssim \frac{a_\star \sqrt{N}}{\eta}, \quad T_2 \lesssim \frac{2}{\eta a_\star} \log\Big(\frac{d}{\varepsilon}\Big). \tag{B.6}$$

*On the same event, with the optimal choice of $a_\star$, the total number of steps is $O\left((Nd)^{1/4} \log\left(\frac{d}{\varepsilon}\right) \log^{1/2}\left(\frac{d}{\delta}\right)\right)$.*

## B.2  Analysis of Phase 1

First we show that a smaller dataset size $N$ leads to faster growth of the outer weight.

**Theorem 3** (Phase 1 is faster with fewer fixed-batch samples).  *Initialize $w^{(0)} \sim \mathrm{Unif}(\mathbb{S}^{d-1})$, and $a^{(0)} \sim \mathcal{N}(0, 1/m)$ where $m$ can be considered as a model width parameter. Fix any target $a_\star \in (0,1]$ and learning rate $\eta > 0$. Consider phase 1 training on a fixed batch of size $N$ and let*

$$T_\star(N) := \min\{t : |a^{(t)}| \geq a_\star\}.$$

*For some choice of $c_0 \in (0, \frac{\sqrt{3}}{2})$, define the event*

$$\mathcal{G}_0 := \Big\{ \mathrm{sign}(a^{(0)}) = \mathrm{sign}(q^{(0)}) \text{ and } |q^{(0)}| \geq c_0/\sqrt{N} \Big\},$$

*and $p_{\mathrm{good}} := \frac{1}{2} p_{PZ}(c_0)$ where $p_{PZ}(c_0)$ is the constant from Lemma 1. Then $\mathbf{Pr}[\mathcal{G}_0] \geq p_{\mathrm{good}}$. On the intersection of $\mathcal{G}_0$ with the stability event $\{\eta a_\star \|\widehat{M}\|_2 \leq 1/2\}$,*

$$T_\star(N) \leq \left\lceil \frac{2(a_\star - |a^{(0)}|)_+}{\eta\, c_0} \sqrt{N} \right\rceil = O\Big(\frac{a_\star}{\eta}\sqrt{N}\Big). \tag{B.7}$$

---

[5] $d \geq 3$ is required for the proof of Lemma 10 regarding the probability of the Beta distribution.
[6] $p_{\mathrm{all}}$ is formally defined in Lemma 10.

*In particular, for every $\delta \in (0,1)$, if $a_\star \leq \min\left\{1, \frac{1}{2\eta B_{N,d,\delta}}\right\}$ with $B_{N,d,\delta}$ from Corollary 4, then equation B.7 holds with probability at least $p_{\text{good}} - \delta$. In particular, holding $(\eta, a_\star)$ fixed, the required number of phase 1 steps scales as $\sqrt{N}$.*

*Proof sketch.* The proof consists of three parts.

1. **Initialization gives a nontrivial $q^{(0)}$ at constant probability.** For $w^{(0)}$ randomly sampled from the unit sphere and an i.i.d. batch of size $N$, the empirical quadratic form $q^{(0)} = (w^{(0)})^\top \widehat{M} w^{(0)} = \frac{1}{N} \sum_{s=1}^N y^{(s)} (x^{(s)\top} w^{(0)})^2$ has magnitude $\Omega(1/\sqrt{N})$ with constant probability (Lemma 1). Since $a^{(0)}$ is initialized symmetric about 0 and independent of $q^{(0)}$, we also have $\mathbf{Pr}[\text{sign}(a^{(0)}) = \text{sign}(q^{(0)})] = 1/2$ conditional on $q^{(0)} \neq 0$ (Lemma 2). Together this yields the constant-probability "good" event $\mathcal{G}_0 = \{\text{sign}(a^{(0)}) = \text{sign}(q^{(0)}), |q^{(0)}| \geq c_0/\sqrt{N}\}$.

2. **While $|a^{(t)}| \leq a_\star$, the sign of $q^{(t)}$ is stable and $|q^{(t)}|$ does not decrease.** Under the stability condition $\eta a_\star \|\widehat{M}\|_2 \leq 1/2$, the normalized updates to the inner weights is a signed power iteration. Lemma 3 shows that the one-step increment $q^{(t+1)} - q^{(t)}$ has the same sign as $a^{(t)}$. On $\mathcal{G}_0$, the outer weight update (with clipping being inactive since $|a^{(t)}|$ hasn't grown to $a_\star < 1$) is $a^{(t+1)} = a^{(t)} + \frac{\eta}{2} q^{(t)}$, hence $\text{sign}(a^{(t)})$ cannot flip as long as $\text{sign}(a^{(t)}) = \text{sign}(q^{(t)})$. Combining these gives by induction that for all $t < T_\star$, $\text{sign}(a^{(t)}) = \text{sign}(q^{(t)}) = \text{sign}(q^{(0)})$ and therefore $|q^{(t)}| \geq |q^{(0)}|$.

3. **Linear growth of $|a^{(t)}|$ and the $\sqrt{N}$ time scale.** On the same event, for all $t < T_\star$ we have $|a^{(t+1)}| = |a^{(t)}| + \frac{\eta}{2}|q^{(t)}| \geq |a^{(t)}| + \frac{\eta}{2}|q^{(0)}|$, so $|a^{(t)}|$ grows at least linearly until it reaches $a_\star$. Thus

$$T_\star \leq \left\lceil \frac{2(a_\star - |a^{(0)}|)_+}{\eta \, |q^{(0)}|} \right\rceil \leq \left\lceil \frac{2(a_\star - |a^{(0)}|)_+}{\eta \, c_0} \sqrt{N} \right\rceil$$

on $\mathcal{G}_0$, which is the claimed $O(\sqrt{N})$ bound.

Finally, Lemma 5 provides a high-probability bound on $\|\widehat{M}\|_2$, yielding an explicit stable choice of $a_\star$ (Corollary 4). $\square$

### B.2.1 CONSTANT-PROBABILITY LOWER BOUND ON $|q^{(0)}| = \Omega(1/\sqrt{N})$

We now prove that, in the parity setting, the fixed-batch quadratic form at initialization

$$q^{(0)} = (w^{(0)})^\top \widehat{M} w^{(0)} = \frac{1}{N} \sum_{s=1}^N y^{(s)} \big(x^{(s)\top} w^{(0)}\big)^2$$

typically has magnitude $\Omega(1/\sqrt{N})$ with *constant* probability.

**Lemma 1.** *Assume $w^{(0)}$ is uniform on the unit sphere (equivalently, $w^{(0)} = g/\|g\|_2$ for $g \sim \mathcal{N}(0, I)$). For some $c_0 \in (0, \frac{\sqrt{3}}{2})$ and $p_{PZ}(c) := (1 - 1/\sqrt{2}) \cdot \frac{(1 - \frac{4}{3}c^2)^2}{3^8}$, for all $N \geq 1$ and all $d \geq 3$,*

$$\mathbf{Pr}\left[|q^{(0)}| \geq \frac{c_0}{\sqrt{N}}\right] \geq p_{PZ}(c_0).$$

*As an example, we can choose $c_0 = \sqrt{3/8}$, in which case $p_{PZ}(c_0) = \frac{2 - \sqrt{2}}{8 \cdot 3^8}$.*

*Proof.* Fix $w = w^{(0)}$ and define a single-sample random variable $Z := y(x^\top w)^2$ so that $q^{(0)} = \frac{1}{N} \sum_{s=1}^N Z_s$ for i.i.d. copies $Z_s$.

**Step 1: conditional mean.**

$$\mu(w) := \mathbb{E}[Z \mid w] = \mathbb{E}\left[x_1 x_2 \Big(\sum_{i=1}^d w_i x_i\Big)^2\right] = \sum_{i,j} w_i w_j \mathbb{E}[x_1 x_2 x_i x_j] = 2 w_1 w_2.$$

**Step 2: conditional variance is bounded below on a constant-probability event over $w$.** Since $y^2 \equiv 1$,

$$\mathbb{E}[Z^2 \mid w] = \mathbb{E}\big[(x^\top w)^4 \mid w\big] = 3\|w\|_2^4 - 2\sum_{i=1}^d w_i^4 \geq \|w\|_2^4 = 1.$$

Therefore,

$$\mathrm{Var}(Z \mid w) = \mathbb{E}[Z^2 \mid w] - \mu(w)^2 \geq 1 - 4w_1^2 w_2^2 \geq 1 - (w_1^2 + w_2^2)^2.$$

Define the event $\mathcal{E} := \{w_1^2 + w_2^2 \leq 1/2\}$. On $\mathcal{E}$ we have $\mathrm{Var}(Z \mid w) \geq 1 - 1/4 = 3/4$. Moreover, since $w$ is uniform on the sphere and $d \geq 3$, the random variable $w_1^2 + w_2^2$ has a $\mathrm{Beta}(1, (d-2)/2)$ distribution, hence

$$\mathbf{Pr}[\mathcal{E}] = \mathbf{Pr}[w_1^2 + w_2^2 \leq 1/2] = 1 - (1 - 1/2)^{(d-2)/2} \geq 1 - 2^{-1/2} =: p_1,$$

where $p_1 > 0$ is an absolute constant.

**Step 3: Paley-Zygmund on $(q^{(0)})^2$.** Condition on $w \in \mathcal{E}$. We already have $\sigma^2(w) := \mathrm{Var}(Z \mid w) \geq 3/4$, hence

$$\mathbb{E}[(q^{(0)})^2 \mid w] = \mu(w)^2 + \frac{\sigma^2(w)}{N} \geq \frac{3}{4N}.$$

We also need to upper bound the second moment of $(q^{(0)})^2$, i.e. a fourth-moment upper bound for $q^{(0)}$. For fixed $w$, the random variable $q^{(0)}$ is a polynomial of total degree at most 4 in the independent Rademacher variables $\{x_i^{(s)}\}_{i \in [d], s \in [N]}$ (after multilinearization using $x_i^2 \equiv 1$). By the Bonami-Beckner (hypercontractive) inequality, for any degree-$d$ polynomial $f$ of Rademachers,

$$(\mathbb{E}[|f|^4])^{1/4} \leq (4-1)^{d/2} (\mathbb{E}[|f|^2])^{1/2}$$

Applying this with $f = q^{(0)}$ (conditional on $w$) gives

$$\mathbb{E}[(q^{(0)})^4 \mid w] \leq 3^8 \mathbb{E}[(q^{(0)})^2 \mid w]^2. \tag{B.8}$$

Apply Paley-Zygmund to the non-negative random variable $Y := (q^{(0)})^2$ conditional on $w \in \mathcal{E}$:

$$\mathbf{Pr}\left[(q^{(0)})^2 \geq \theta \mathbb{E}[(q^{(0)})^2 \mid w] \,\Big|\, w\right] \geq \frac{(1-\theta)^2 \mathbb{E}[(q^{(0)})^2 \mid w]^2}{\mathbb{E}[(q^{(0)})^4 \mid w]} \geq \frac{(1-\theta)^2}{3^8},$$

where we used equation B.8. With $\theta = 1/2$,

$$\mathbf{Pr}\left[(q^{(0)})^2 \geq \theta \mathbb{E}[(q^{(0)})^2 \mid w] \,\Big|\, w\right] \geq \frac{1}{4 \cdot 3^8} =: p_2$$

On this event,

$$|q^{(0)}| \geq \sqrt{\tfrac{1}{2}\mathbb{E}[(q^{(0)})^2 \mid w]} \geq \sqrt{\frac{3}{8}} \cdot \frac{1}{\sqrt{N}}.$$

Thus, for $c_0 := \sqrt{3/8}$,

$$\mathbf{Pr}\left[|q^{(0)}| \geq \frac{c_0}{\sqrt{N}}\right] \geq \mathbf{Pr}[\mathcal{E}] \cdot p_2 \geq p_1 p_2 =: p_{ZL}(c_0).$$

$\square$

We also need the lucky event of $\mathrm{sign}(a^{(0)}) = \mathrm{sign}(q^{(0)})$ so that the updates don't flip the output weight's sign.

**Lemma 2.** *Assume $a^{(0)}$ is independent of $(w^{(0)}, \widehat{M})$, symmetric about 0, and that $\mathbf{Pr}[a^{(0)} = 0] = 0$. Then for every threshold $q_\star > 0$,*

$$\mathbf{Pr}\left[\mathrm{sign}(a^{(0)}) = \mathrm{sign}(q^{(0)}) \text{ and } |q^{(0)}| \geq q_\star\right] = \frac{1}{2}\mathbf{Pr}\left[|q^{(0)}| \geq q_\star\right].$$

*Proof.* Condition on $(w^{(0)}, \widehat{M})$ so that $q^{(0)}$ is fixed. On the event $q^{(0)} \neq 0$, symmetry and independence of $a^{(0)}$ imply $\mathbf{Pr}[\mathrm{sign}(a^{(0)}) = \mathrm{sign}(q^{(0)}) \mid q^{(0)}] = 1/2$. Multiply by $\mathbb{1}\{|q^{(0)}| \geq q_\star\}$ and average over $(w^{(0)}, \widehat{M})$. $\square$

B.2.2   STABILITY OF $q^{(t)}$ FOR SMALL $a^{(t)}$

Next, we show that under a stability condition of $\eta|a|\|\widehat{\boldsymbol{M}}\|_2 \leq 1/2$, the update in $q$ has the same sign as $a$.

**Lemma 3.** *Fix any unit vector $w \in \mathbb{R}^d$, scalar $a \in \mathbb{R}$, and learning rate $\eta > 0$. Define*

$$\tilde{w} := (\boldsymbol{I} + \eta a \widehat{\boldsymbol{M}})w, \qquad w^+ := \tilde{w}/\|\tilde{w}\|_2.$$

*Let $q := w^\top \widehat{\boldsymbol{M}} w$ as before, and define $q^+ := (w^+)^\top \widehat{\boldsymbol{M}} w^+$ similarly.*

*Then, under the stability condition of $\eta|a|\|\widehat{\boldsymbol{M}}\|_2 \leq 1/2$, $q^+ - q$ has the same sign as $a$ (or is zero).*

*Proof.* Define the following quantities

$$s := w^\top \widehat{\boldsymbol{M}}^2 w, \qquad r := w^\top \widehat{\boldsymbol{M}}^3 w. \tag{B.9}$$

With $\widehat{\boldsymbol{M}}$ being symmetric, we have

$$q^+ = \frac{\tilde{w}^\top \widehat{\boldsymbol{M}} \tilde{w}}{\tilde{w}^\top \tilde{w}} = \frac{w^\top (\widehat{\boldsymbol{M}} + 2\eta a \widehat{\boldsymbol{M}}^2 + \eta^2 a^2 \widehat{\boldsymbol{M}}^3)w}{w^\top (\boldsymbol{I} + 2\eta a \widehat{\boldsymbol{M}} + \eta^2 a^2 \widehat{\boldsymbol{M}}^2)w} = \frac{q + 2\eta a s + \eta^2 a^2 r}{1 + 2\eta a q + \eta^2 a^2 s},$$

and the update in $q$ is

$$q^+ - q = \frac{2\eta a(s - q^2) + \eta^2 a^2(r - qs)}{1 + 2\eta a q + \eta^2 a^2 s}. \tag{B.10}$$

Note that the denominator in equation B.10 is positive, which follows from the stability assumption $\eta|a|\|\widehat{\boldsymbol{M}}\|_2 \leq 1/2$. The sign of $q^+ - q$ hence depends on the two terms in the numerator, which we bound separately below.

First, let $\widehat{\boldsymbol{M}} = \sum_{i=1}^d \lambda_i u_i u_i^\top$ be an eigendecomposition and set $\alpha_i := \langle w, u_i\rangle^2$ so that $\alpha$ is a probability vector. Let $\Lambda$ be the random variable taking value $\lambda_i$ with probability $\alpha_i$. Then

$$q = \sum_i \alpha_i \lambda_i = \mathbb{E}[\Lambda], \qquad s = \sum_i \alpha_i \lambda_i^2 = \mathbb{E}[\Lambda^2], \qquad r = \sum_i \alpha_i \lambda_i^3.$$

This directly gives that $s - q^2 = \mathrm{Var}(\Lambda) \geq 0$.

For the second term, note that

$$r - qs = \mathbb{E}[\Lambda^3] - \mathbb{E}[\Lambda]\mathbb{E}[\Lambda^2] = \mathbb{E}\big[(\Lambda - \mathbb{E}[\Lambda])^2(\Lambda + \mathbb{E}[\Lambda])\big].$$

Since $|\Lambda + \mathbb{E}[\Lambda]| \leq 2\|\widehat{\boldsymbol{M}}\|_2$, we obtain

$$|r - qs| \leq 2\|\widehat{\boldsymbol{M}}\|_2 \mathbb{E}[(\Lambda - \mathbb{E}[\Lambda])^2] = 2\|\widehat{\boldsymbol{M}}\|_2 (s - q^2),$$

Combining this with the stability assumption of $\eta|a|\|\widehat{\boldsymbol{M}}\|_2 \leq 1/2$, we can bound the second term of the numerator in Equation (B.10) by

$$\left|\eta^2 a^2(r - qs)\right| \leq 2\eta^2 |a|^2 \|\widehat{\boldsymbol{M}}\|_2 (s - q^2) \leq \eta|a|(s - q^2).$$

Therefore,

- if $a \geq 0$, then $2\eta a(s - q^2) + \eta^2 a^2(r - qs) \geq 2\eta a(s - q^2) - \eta a(s - q^2) = \eta a(s - q^2) \geq 0$;

- if $a \leq 0$, then $2\eta a(s - q^2) + \eta^2 a^2(r - qs) \leq 2\eta a(s - q^2) + \eta|a|(s - q^2) = \eta a(s - q^2) \leq 0$.

Thus the numerator in equation B.10 and hence $q^+ - q$ has the same sign as $a$ (or is zero). $\qquad\square$

### B.2.3 LINEAR GROWTH OF $a^{(t)}$

Next, we show that $a$ grows linearly when conditioned on the lucky event in Lemma 2 and the stability assumption in Lemma 3, from which an upper bound on $T_\star$ (i.e., time for $a$ to grow to $a_\star$) directly follows.

**Lemma 4.** *Assume the initialization event*

$$\mathrm{sign}(a^{(0)}) = \mathrm{sign}(q^{(0)}) \qquad and \qquad |q^{(0)}| \geq q_\star > 0,$$

*for $q_\star := \frac{c_0}{\sqrt{N}}$ from Lemma 1. Further, assume the stability condition*

$$\eta a_\star \|\widehat{M}\|_2 \leq \frac{1}{2}.$$

*Then for all $t < T_\star$ we have $\mathrm{sign}(a^{(t)}) = \mathrm{sign}(q^{(t)}) = \mathrm{sign}(q^{(0)})$ and $|q^{(t)}| \geq |q^{(0)}| \geq q_\star$. Consequently,*

$$T_\star \leq \left\lceil \frac{2(a_\star - |a^{(0)}|)_+}{\eta q_\star} \right\rceil. \tag{B.11}$$

*Proof.* Fix any $t < T_\star$. Since $|a^{(t)}| \leq a_\star$ and $\eta a_\star \|\widehat{M}\|_2 \leq 1/2$, we may apply Lemma 3 to the inner weight update at time $t$. It implies that $q^{(t+1)} - q^{(t)}$ has the same sign as $a^{(t)}$ (or is zero).

We next show by induction that $\mathrm{sign}(a^{(t)}) = \mathrm{sign}(q^{(t)}) = \mathrm{sign}(q^{(0)})$ and $|q^{(t)}| \geq |q^{(0)}|$ for all $t < T_\star$. The base case $t = 0$ holds by assumption. Assume it holds at time $t$. Because $t < T_\star$ we have $|a^{(t)}| < 1$, so clipping is inactive and

$$a^{(t+1)} = a^{(t)} + \frac{\eta}{2} q^{(t)}.$$

Since $\mathrm{sign}(a^{(t)}) = \mathrm{sign}(q^{(t)})$, we get $\mathrm{sign}(a^{(t+1)}) = \mathrm{sign}(a^{(t)})$ and

$$|a^{(t+1)}| = |a^{(t)}| + \frac{\eta}{2}|q^{(t)}|.$$

Thus $\mathrm{sign}(a^{(t)})$ remains constant and equal to $\mathrm{sign}(q^{(0)})$ throughout $t < T_\star$. Returning to Lemma 3, this means $q^{(t)}$ is pushed monotonically in the direction of $\mathrm{sign}(a^{(t)}) = \mathrm{sign}(q^{(0)})$ and therefore cannot cross 0. Hence $\mathrm{sign}(q^{(t)}) = \mathrm{sign}(q^{(0)})$ and $|q^{(t)}| \geq |q^{(0)}| \geq q_\star$ for all $t < T_\star$.

Finally, using $|q^{(t)}| \geq q_\star$ gives the linear growth bound

$$|a^{(t+1)}| = |a^{(t)}| + \frac{\eta}{2}|q^{(t)}| \geq |a^{(t)}| + \frac{\eta}{2} q_\star,$$

so $|a^{(t)}| \geq |a^{(0)}| + t \cdot \frac{\eta}{2} q_\star$ while $t < T_\star$. Solving for the first $t$ such that $|a^{(t)}| \geq a_\star$ yields equation B.11. $\qquad\square$

### B.2.4 CHOOSING LARGEST STABLE $a_\star$

In order to find a bound on how large we can set $a_\star$, we will first bound $\|\widehat{M}\|_2$.

**Lemma 5** (Matrix Bernstein bound for $\|\widehat{M}\|_2$)**.** *For every $\delta \in (0,1)$, with probability at least $1 - \delta$,*

$$\|\widehat{M}\|_2 \leq 1 + C\left(\sqrt{\frac{d\log(2d/\delta)}{N}} + \frac{d\log(2d/\delta)}{N}\right)$$

*for a universal constant $C > 0$.*

*Proof.* Let $A_s := y^{(s)} x^{(s)} x^{(s)\top}$. Since $A_s^2 = (x^{(s)} x^{(s)\top})^2 = \|x^{(s)}\|_2^2 x^{(s)} x^{(s)\top} = d x^{(s)} x^{(s)\top}$ (and $y^{(s)2} = 1$), we have

$$\mathbb{E}[A_s^2] = d\mathbb{E}[xx^\top] = d\boldsymbol{I}.$$

Write the population matrix as

$$\boldsymbol{M} := \mathbb{E}[A_s] = \mathbb{E}[yxx^\top] = e_1 e_2^\top + e_2 e_1^\top, \qquad \|\boldsymbol{M}\|_2 = 1.$$

Define centered summands $X_s := A_s - \boldsymbol{M}$ so that $\mathbb{E}[X_s] = 0$ and

$$\widehat{\boldsymbol{M}} - \boldsymbol{M} = \frac{1}{N} \sum_{s=1}^{N} X_s.$$

We bound $\|X_s\|_2 \le \|A_s\|_2 + \|\boldsymbol{M}\|_2 \le d + 1 \le 2d$, so we may take $R := 2d$.

$$\mathbb{E}[X_s^2] = \mathbb{E}[(A_s - \boldsymbol{M})^2] = \mathbb{E}[A_s^2] - \boldsymbol{M}^2 \preceq \mathbb{E}[A_s^2] = d\boldsymbol{I},$$

hence

$$\sigma^2 := \left\| \sum_{s=1}^{N} \mathbb{E}[X_s^2] \right\|_2 \le Nd.$$

Matrix Bernstein (for sums of independent mean-zero self-adjoint matrices) then yields that with probability at least $1 - \delta$,

$$\left\| \sum_{s=1}^{N} X_s \right\|_2 \le C \left( \sqrt{\sigma^2 \log(2d/\delta)} + R \log(2d/\delta) \right) \le C \left( \sqrt{Nd \log(2d/\delta)} + d \log(2d/\delta) \right).$$

Dividing by $N$ gives

$$\|\widehat{\boldsymbol{M}} - \boldsymbol{M}\|_2 \le C \left( \sqrt{\frac{d \log(2d/\delta)}{N}} + \frac{d \log(2d/\delta)}{N} \right).$$

Finally, $\|\widehat{\boldsymbol{M}}\|_2 \le \|\boldsymbol{M}\|_2 + \|\widehat{\boldsymbol{M}} - \boldsymbol{M}\|_2$ and $\|\boldsymbol{M}\|_2 = 1$, proving the claim. $\qquad \square$

Substituting the above gives the following.

**Corollary 4.** *Fix $\delta \in (0, 1)$ and stepsize $\eta > 0$. Let $B_{N,d,\delta} = 1 + C \left( \sqrt{\frac{d \log(2d/\delta)}{N}} + \frac{d \log(2d/\delta)}{N} \right)$. If*

$$a_\star \le \min \left\{ 1, \frac{1}{2\eta B_{N,d,\delta}} \right\},$$

*then with probability at least $1 - \delta$ the stability event*

$$\eta |a^{(t)}| \|\widehat{\boldsymbol{M}}\|_2 \le \tfrac{1}{2} \qquad \text{for all } t < T_\star := \min\{t : |a^{(t)}| \ge a_\star\}$$

*holds.*

### B.3   Analysis of Phase 2

In Phase 2, we replace $\widehat{\boldsymbol{M}}$ by the population matrix $\boldsymbol{M} = e_1 e_2^\top + e_2 e_1^\top$. Its spectrum is explicit: let

$$u_+ := \frac{e_1 + e_2}{\sqrt{2}}, \qquad u_- := \frac{e_1 - e_2}{\sqrt{2}},$$

then $\boldsymbol{M} u_+ = u_+$, $\boldsymbol{M} u_- = -u_-$, and $\boldsymbol{M} v = 0$ for all $v \perp \text{span}\{e_1, e_2\}$.

We show that in Phase 2, $w$ converges quickly to one of $u_+, u_-$ following a power iteration on $\boldsymbol{M}$. [7]

---

[7]The upper bound on $T_2$ is likely improvable to $O(\log(1/a^{(0)}))$.

**Lemma 6** (Population contraction). *Assume $\eta \leq \frac{1}{2}$, $a^{(0)} \neq 0$, and $\mathrm{sign}(a^{(0)}) = \mathrm{sign}(q^{(0)})$. Consider projected updates*

$$a^{(t+1)} = \mathrm{clip}_{[-1,1]}\left(a^{(t)} + \tfrac{\eta}{2}q^{(t)}\right), \qquad w^{(t+1)} = \frac{w^{(t)} + \eta a^{(t)} \boldsymbol{M} w^{(t)}}{\|w^{(t)} + \eta a^{(t)} \boldsymbol{M} w^{(t)}\|_2}, \qquad q^{(t)} := (w^{(t)})^\top \boldsymbol{M} w^{(t)}.$$

*Let*

$$u_a := \frac{e_1 + \mathrm{sign}(a^{(0)})e_2}{\sqrt{2}}, \quad u_{-a} := \frac{e_1 - \mathrm{sign}(a^{(0)})e_2}{\sqrt{2}}, \qquad \alpha_t := |\langle w^{(t)}, u_a \rangle|, \qquad r_t := \frac{\sqrt{1 - \alpha_t^2}}{\alpha_t}.$$

*Then:*

1. **Sign stability and monotonicity.** *For all $t \geq 0$, $\mathrm{sign}(a^{(t)}) = \mathrm{sign}(q^{(t)}) = \mathrm{sign}(a^{(0)})$, and $|a^{(t)}|$ is non-decreasing.*

2. **Alignment contraction.** *For all $t \geq 0$,*

$$r_{t+1} \leq \frac{1}{1 + \eta|a^{(t)}|} r_t \leq \frac{1}{1 + \eta|a^{(0)}|} r_t.$$

*Consequently, after*

$$T_2 := \left\lceil \frac{2}{\eta|a^{(0)}|} \log\left(\frac{1}{\alpha_0^2 \varepsilon}\right) \right\rceil$$

*steps we have $\alpha_{T_2}^2 \geq 1 - \varepsilon$ for any $\varepsilon \in (0, 1/2)$.*

*Proof.* Because $\eta \leq 1/2$ and $|a^{(t)}| \leq 1$, the stability condition of Lemma 3 (i.e., $\eta|a^{(t)}|\|\boldsymbol{M}\|_2 \leq \frac{1}{2}$) holds with $\widehat{\boldsymbol{M}} = \boldsymbol{M}$ for every step. Hence $q^{(t+1)} - q^{(t)}$ has the same sign as $a^{(t)}$ (or is 0). Since $\mathrm{sign}(a^{(0)}) = \mathrm{sign}(q^{(0)})$ and

$$a^{(t+1)} = a^{(t)} + \tfrac{\eta}{2}q^{(t)} \quad \text{as long as clipping is inactive,}$$

the signs of $a^{(t)}$ and $q^{(t)}$ cannot flip; moreover $|a^{(t)}|$ is nondecreasing, and clipping preserves the sign once $|a^{(t)}|$ hits 1. This proves (1).

For (2), decompose $w^{(t)}$ as

$$w^{(t)} = c_t u_a + b_t u_{-a} + v_t,$$

where $v_t \perp \mathrm{span}\{e_1, e_2\}$. Then $(\boldsymbol{I} + \eta a \boldsymbol{M})u_a = (1 + \eta|a|)u_a$, $(\boldsymbol{I} + \eta a \boldsymbol{M})u_{-a} = (1 - \eta|a|)u_{-a}$, and $(\boldsymbol{I} + \eta a \boldsymbol{M})v = v$. Thus before normalization,

$$(\boldsymbol{I} + \eta a \boldsymbol{M})w^{(t)} = (1 + \eta|a|)c_t u_a + (1 - \eta|a|)b_t u_{-a} + v_t.$$

After normalization, ratios between the orthogonal component and the $u_a$ component is

$$r_{t+1} = \frac{\sqrt{b_{t+1}^2 + \|v_{t+1}\|^2}}{|c_{t+1}|} = \frac{\sqrt{(1 - \eta|a^{(t)}|)^2 b_t^2 + \|v_t\|^2}}{(1 + \eta|a^{(t)}|)|c_t|} \leq \frac{\sqrt{b_t^2 + \|v_t\|^2}}{(1 + \eta|a^{(t)}|)|c_t|} = \frac{r_t}{1 + \eta|a^{(t)}|}. \tag{B.12}$$

Combined with (1), this shows that $r_t$ contracts by at least a factor of $\frac{1}{1+\eta|a^{(0)}|}$, which is strictly smaller than 1 since $a^{(0)} \neq 0$.

The convergence time $T_2$ follows from $\log(1 + \eta|a|) \geq \frac{\eta|a|}{1+\eta|a|} \geq \frac{\eta|a|}{2}$, since $\eta|a| \leq 1$.

$\square$

## B.4 Combining both phases

We have shown that $T_\star \leq \left\lceil \frac{2(a_\star - |a^{(0)}|)}{\eta q_\star} \right\rceil$ (Lemma 4) and $T_2 \leq \frac{2}{\eta|a_\star|} \log\left(\frac{1}{\alpha_0^2 \varepsilon}\right)$ (Lemma 6). To reason about the overall time $T_\star + T_2$, it remains to check how $\alpha_0$ depends on $a_\star$, which in turn depends on how much $w$ moves during Phase 1.

In the following, we will first bound $w$'s drift (Lemma 7) which will then allow us to relate $a_\star$ and $\alpha_0$ (Lemma 8), and present the final convergence bound in Appendix B.4.3.

### B.4.1  $w^{(t)}$ GROWS SLOWLY

We first need to bound how much $w$ drifts in phase 1. We show that under the assumptions of Lemma 4, the input weight $w^{(t)}$ changes little up to time $T_\star$.

**Lemma 7** (Control of inner weight drift up to $T_\star$). *Under the assumptions of Lemma 4,*

$$\|w^{(T_\star)} - w^{(0)}\|_2 \leq \frac{8\|\widehat{M}\|_2}{q_\star} a_\star(a_\star - |a^{(0)}|)_+ \ + \ 4\eta\|\widehat{M}\|_2 a_\star. \tag{B.13}$$

*Proof.* Write the pre-normalization iterate as

$$\tilde{w}^{(t+1)} = w^{(t)} + \eta a^{(t)}\widehat{M}w^{(t)}, \qquad w^{(t+1)} = \tilde{w}^{(t+1)}/\|\tilde{w}^{(t+1)}\|_2.$$

Let $u^{(t)} := \eta a^{(t)}\widehat{M}w^{(t)}$, so $\tilde{w}^{(t+1)} = w^{(t)} + u^{(t)}$. For $t < T_\star$ we have $|a^{(t)}| \leq a_\star$, hence

$$\|u^{(t)}\|_2 \leq \eta|a^{(t)}|\|\widehat{M}\|_2 \leq \eta a_\star\|\widehat{M}\|_2 \leq \frac{1}{2}.$$

For any unit vector $w$ and any $u$ with $\|u\|_2 \leq 1/2$, one has the standard normalization Lipschitz bound

$$\left\|\frac{w+u}{\|w+u\|_2} - w\right\|_2 \leq 4\|u\|_2,$$

which we apply with $(w, u) = (w^{(t)}, u^{(t)})$ to get

$$\|w^{(t+1)} - w^{(t)}\|_2 \leq 4\|u^{(t)}\|_2 \leq 4\eta|a^{(t)}|\|\widehat{M}\|_2.$$

Summing over $t = 0, 1, \ldots, T_\star - 1$ yields

$$\|w^{(T_\star)} - w^{(0)}\|_2 \leq 4\eta\|\widehat{M}\|_2 \sum_{t < T_\star} |a^{(t)}|.$$

Using the crude bound $\sum_{t < T_\star} |a^{(t)}| \leq T_\star a_\star$, which is sufficient for the final scaling, together with equation B.11 gives

$$\sum_{t < T_\star} |a^{(t)}| \leq a_\star\left(\frac{2(a_\star - |a^{(0)}|)_+}{\eta q_\star} + 1\right) = \frac{2a_\star(a_\star - |a^{(0)}|)_+}{\eta q_\star} + a_\star.$$

Plugging in yields

$$\|w^{(T_\star)} - w^{(0)}\|_2 \leq 4\eta\|\widehat{M}\|_2\left(\frac{2a_\star(a_\star - |a^{(0)}|)_+}{\eta q_\star} + a_\star\right) = \frac{8\|\widehat{M}\|_2}{q_\star} a_\star(a_\star - |a^{(0)}|)_+ + 4\eta\|\widehat{M}\|_2 a_\star,$$

which is equation B.13. $\qquad\square$

### B.4.2  CONNECTING $\alpha_0$ AND $a_\star$

**Lemma 8** (Lower bound on $\alpha_0^2$ in terms of $a_\star$). *Let $u := (e_1 + \text{sign}(a^{(T_\star)})e_2)/\sqrt{2}$ and define*

$$\alpha_0 := |\langle w^{(T_\star)}, u\rangle|.$$

*On the event $\|w^{(T_\star)} - w^{(0)}\|_2 \leq \varepsilon_{\text{drift}}$, we have*

$$\alpha_0^2 \geq \left(|\langle w^{(0)}, u\rangle| - \varepsilon_{\text{drift}}\right)_+^2.$$

*Under the assumptions of Lemma 7, we may take*

$$\varepsilon_{\text{drift}} := \frac{8\|\widehat{M}\|_2}{q_\star} a_\star(a_\star - |a^{(0)}|)_+ \ + \ 4\eta\|\widehat{M}\|_2 a_\star,$$

*so $\alpha_0^2$ is explicitly lower bounded in terms of $a_\star$.*

*Proof.* By Cauchy–Schwarz, $|\langle w^{(T_\star)}, u\rangle - \langle w^{(0)}, u\rangle| \leq \|w^{(T_\star)} - w^{(0)}\|_2 \|u\|_2 = \|w^{(T_\star)} - w^{(0)}\|_2$. This implies $|\langle w^{(T_\star)}, u\rangle| \geq |\langle w^{(0)}, u\rangle| - \varepsilon_{\text{drift}}$ on the event. The explicit choice of $\varepsilon_{\text{drift}}$ is equation B.13. $\qquad\square$

**Lemma 9** (Constant-probability lower bound on random initialization alignment). *Let $u \in \mathbb{R}^d$ be any fixed unit vector and let $w^{(0)}$ be uniform on the unit sphere. Then there exists a universal constant $p_{\text{align}} > 0$ such that for all $d \geq 2$,*

$$\mathbf{Pr}\left[|\langle w^{(0)}, u\rangle| \geq \frac{1}{2\sqrt{d}}\right] \geq p_{\text{align}}.$$

*Proof.* Write $w^{(0)} = g/\|g\|_2$ for $g \sim \mathcal{N}(0, \boldsymbol{I})$ and rotate so that $u = e_1$. Then $|\langle w^{(0)}, u\rangle| = |g_1|/\|g\|_2$. On the event $\{|g_1| \geq 1\} \cap \{\|g\|_2 \leq 2\sqrt{d}\}$ we have $|g_1|/\|g\|_2 \geq 1/(2\sqrt{d})$. $\mathbf{Pr}\left[\{|g_1| \geq 1\} \cap \{\|g\|_2 \leq 2\sqrt{d}\}\right] \geq \mathbf{Pr}\left[|g_1| \geq 1\right] - \mathbf{Pr}\left[\|g\|_2 > 2\sqrt{d}\right]$. The former one has constant probability and the latter one decays exponentially with $d$, so the intersection has probability at least some universal constant $p_{\text{align}} > 0$. $\qquad\square$

**Lemma 10** (Constant-probability simultaneous phase-1 bootstrap and population sign alignment). *Let*

$$u_\pm := \frac{e_1 \pm e_2}{\sqrt{2}}, \qquad P_{12} := u_+ u_+^\top + u_- u_-^\top, \qquad q_{\text{pop}}(w) := w^\top \boldsymbol{M} w.$$

*There exist universal constants $c_0 > 0$ and $p_{\text{all}} > 0$ such that, with*

$$\mathcal{P}_0 := \left\{|q_{\text{pop}}(w^{(0)})| \geq \frac{3}{4d}, \quad \|P_{12} w^{(0)}\|_2 \leq \frac{1}{\sqrt{d}}\right\}$$

*and*

$$\mathcal{G}_{\text{sign}} := \left\{\text{sign}(a^{(0)}) = \text{sign}(q^{(0)}) = \text{sign}(q_{\text{pop}}(w^{(0)})), \quad |q^{(0)}| \geq \frac{c_0}{\sqrt{N}}\right\},$$

*we have*

$$\mathbf{Pr}\left[\mathcal{G}_{\text{sign}} \cap \mathcal{P}_0\right] \geq p_{\text{all}}.$$

*In particular, on this event, the empirical sign used to grow $a$ in phase 1 is already the population sign that will be needed at the start of phase 2.*

*Proof.* Write $z_\pm := \langle w^{(0)}, u_\pm\rangle$ and $r^2 := z_+^2 + z_-^2 = \|P_{12} w^{(0)}\|_2^2$. Conditional on $r$, the angle $(z_+, z_-)/r$ is uniform on the unit circle, and $r^2 \sim \text{Beta}(1, (d-2)/2)$. Consider the event

$$\mathcal{R} := \left\{\frac{9}{10d} \leq r^2 \leq \frac{1}{d}, \qquad |\cos(2\theta)| \geq \frac{5}{6}\right\}, \qquad (z_+, z_-) = r(\cos\theta, \sin\theta).$$

On $\mathcal{R}$,

$$|q_{\text{pop}}(w^{(0)})| = |z_+^2 - z_-^2| = r^2 |\cos(2\theta)| \geq \frac{3}{4d}, \qquad \|P_{12} w^{(0)}\|_2 = r \leq \frac{1}{\sqrt{d}},$$

so $\mathcal{R} \subseteq \mathcal{P}_0$. The radial probability of $\{9/(10d) \leq r^2 \leq 1/d\}$ is bounded below by a universal constant for all $d \geq 3$ after decreasing the constant to cover the finitely many small dimensions, and the angular event $\{|\cos(2\theta)| \geq 5/6\}$ also has universal positive probability. Hence $\mathbf{Pr}[\mathcal{P}_0] \geq p_{\text{pop}} > 0$.

Fix any $w \in \mathcal{P}_0$ and set $\mu := q_{\text{pop}}(w)$. For one phase–1 sample, let $Z := y(x^\top w)^2$, so that $\mathbb{E}[Z \mid w] = \mu$ and $q^{(0)} = N^{-1} \sum_{s=1}^N Z_s$. Since $\|P_{12} w\|_2^2 \leq 1/d \leq 1/2$, the variance lower bound in Lemma 1 gives $\text{Var}(Z \mid w) \geq 3/4$. The same hypercontractive fourth-moment bound used in Lemma 1, applied to $\sqrt{N}(q^{(0)} - \mu)$, gives a universal fourth-moment upper bound.

We use the following elementary one-sided consequence of these two moment bounds: if $X$ is mean zero, $\mathbb{E}[X^2] = \sigma^2$, and $\mathbb{E}[X^4] \leq K\sigma^4$, then there are constants $c_K, p_K > 0$ depending only on $K$ such that $\mathbf{Pr}[X \geq c_K \sigma] \geq p_K$. Indeed,

writing $X_+ = \max\{X, 0\}$ and $X_- = \max\{-X, 0\}$, the identity $\mathbb{E}[X_+] = \mathbb{E}[X_-]$ and interpolation between $L_1, L_2, L_4$ norms give $\mathbb{E}[X_+] \geq \sigma/(2^{3/2}\sqrt{K})$. Therefore, with $\theta := 1/(2^{5/2}\sqrt{K})$,

$$\mathbb{E}[X_+] \leq \theta\sigma + \left(\mathbb{E}[X_+^2]\right)^{1/2} \mathbf{Pr}[X \geq \theta\sigma]^{1/2} \leq \theta\sigma + \sigma\,\mathbf{Pr}[X \geq \theta\sigma]^{1/2},$$

which implies $\mathbf{Pr}[X \geq \theta\sigma] \geq \theta^2$. Thus one may take $c_K = \theta$ and $p_K = \theta^2$. Applying this to $X := \mathrm{sign}(\mu)\sqrt{N}(q^{(0)} - \mu)$, and using $\mathrm{sign}(\mu)\sqrt{N}\,q^{(0)} = X + |\mu|\sqrt{N} \geq X$, gives, after decreasing $c_0$ if necessary, a universal $p_{\mathrm{one}} > 0$ such that

$$\mathbf{Pr}\left[\mathrm{sign}(\mu)\,q^{(0)} \geq \frac{c_0}{\sqrt{N}} \,\middle|\, w^{(0)} = w\right] \geq p_{\mathrm{one}}.$$

Equivalently, conditional on $w \in \mathcal{P}_0$, the empirical quadratic form has the same sign as the population quadratic form and has magnitude at least $c_0/\sqrt{N}$ with probability at least $p_{\mathrm{one}}$. Finally, $a^{(0)}$ is independent of the samples and is symmetric about zero, so conditional on $(w^{(0)}, \widehat{M})$ the event $\mathrm{sign}(a^{(0)}) = \mathrm{sign}(q^{(0)})$ contributes an additional factor $1/2$. Therefore

$$\mathbf{Pr}\left[\mathcal{G}_{\mathrm{sign}} \cap \mathcal{P}_0\right] \geq \frac{1}{2}p_{\mathrm{pop}}p_{\mathrm{one}} := p_{\mathrm{all}} > 0.$$

$\square$

**Lemma 11** (Phase-1 drift preserves the population sign). *Define $q_{\mathrm{pop}}(w) := w^\top M w$ and $P_{12} := u_+ u_+^\top + u_- u_-^\top$ as in Lemma 10. Suppose*

$$|q_{\mathrm{pop}}(w^{(0)})| \geq \frac{3}{4d}, \qquad \|P_{12}w^{(0)}\|_2 \leq \frac{1}{\sqrt{d}}, \qquad \|w^{(T_\star)} - w^{(0)}\|_2 \leq \frac{1}{4\sqrt{d}}.$$

*Then*

$$\mathrm{sign}\left(q_{\mathrm{pop}}(w^{(T_\star)})\right) = \mathrm{sign}\left(q_{\mathrm{pop}}(w^{(0)})\right).$$

*Moreover, if $s := \mathrm{sign}(q_{\mathrm{pop}}(w^{(0)}))$ and $u_s := (e_1 + se_2)/\sqrt{2}$, then*

$$|\langle w^{(T_\star)}, u_s\rangle| \geq \left(\frac{\sqrt{3}}{2} - \frac{1}{4}\right)\frac{1}{\sqrt{d}} \geq \frac{1}{2\sqrt{d}}.$$

*Proof.* Let $\Delta := w^{(T_\star)} - w^{(0)}$. Since $\|M\|_2 = 1$ and $\|Mw^{(0)}\|_2 = \|P_{12}w^{(0)}\|_2$, we have

$$\begin{aligned}
\left|q_{\mathrm{pop}}(w^{(T_\star)}) - q_{\mathrm{pop}}(w^{(0)})\right| &= \left|2(w^{(0)})^\top M\Delta + \Delta^\top M\Delta\right| \\
&\leq 2\|P_{12}w^{(0)}\|_2\|\Delta\|_2 + \|\Delta\|_2^2 \\
&\leq \frac{1}{2d} + \frac{1}{16d} = \frac{9}{16d} < \frac{3}{4d}.
\end{aligned}$$

Thus the perturbation is smaller than the initial population margin, so the sign of $q_{\mathrm{pop}}$ is preserved.

For the alignment claim, write $z_s := \langle w^{(0)}, u_s\rangle$ and $z_{-s} := \langle w^{(0)}, u_{-s}\rangle$. Since $s\,q_{\mathrm{pop}}(w^{(0)}) = z_s^2 - z_{-s}^2 \geq 3/(4d)$, we have $|z_s| \geq \sqrt{3}/(2\sqrt{d})$. Cauchy-Schwarz gives

$$|\langle w^{(T_\star)}, u_s\rangle| \geq |\langle w^{(0)}, u_s\rangle| - \|w^{(T_\star)} - w^{(0)}\|_2 \geq \left(\frac{\sqrt{3}}{2} - \frac{1}{4}\right)\frac{1}{\sqrt{d}} \geq \frac{1}{2\sqrt{d}}.$$

$\square$

### B.4.3  FINAL CONVERGENCE BOUND

Fix $\varepsilon \in (0, 1/2)$ and $\delta \in (0, 1)$. Run phase 1 on a fixed batch of size $N$ using updates equation B.4–equation B.5 until time

$$T_1 := \min\{t : |a^{(t)}| \geq a_\star\}.$$

Let $p_{\text{all}} > 0$ be the universal constant from Lemma 10. Then, by intersecting Lemma 10 with the operator-norm event $\{\|\widehat{M}\|_2 \leq B_{N,d,\delta}\}$, we get

$$\mathbf{Pr}\left[\mathcal{G}_{\text{sign}} \cap \mathcal{P}_0 \cap \{\|\widehat{M}\|_2 \leq B_{N,d,\delta}\}\right] \geq p_{\text{all}} - \delta.$$

On this event, the following hold:

1. **Phase 1 time.** We have

$$T_1 \leq \left\lceil \frac{2(a_\star - |a^{(0)}|)_+}{\eta\, q_\star} \right\rceil \leq \left\lceil \frac{2a_\star}{\eta\, c_0} \sqrt{N} \right\rceil.$$

2. **Phase 2 alignment.** Switch to population gradients ($\widehat{M} \leftarrow M$), and run the population contraction on $w$ for

$$T_2 := \left\lceil \frac{2}{\eta a_\star} \log\left(\frac{16d}{\varepsilon}\right) \right\rceil$$

steps.

By Lemma 7 and the above choice of $a_\star$, we have

$$\|w^{(T_1)} - w^{(0)}\|_2 \leq \varepsilon_{\text{drift}} \leq \frac{1}{4\sqrt{d}}.$$

Since $\mathcal{G}_{\text{sign}}$ gives $\text{sign}(a^{(0)}) = \text{sign}(q_{\text{pop}}(w^{(0)}))$ and phase 1 preserves $\text{sign}(a^{(t)})$ up to $T_1$, Lemma 11 gives the missing population sign condition

$$\text{sign}(a^{(T_1)}) = \text{sign}\left((w^{(T_1)})^\top M w^{(T_1)}\right).$$

The same lemma also gives

$$\alpha_0^2 := \left|\left\langle w^{(T_1)}, \frac{e_1 + \text{sign}(a^{(T_1)})e_2}{\sqrt{2}} \right\rangle\right|^2 \geq \frac{1}{4d}.$$

Therefore Lemma 6, applied from the phase–2 starting point and using $|a^{(T_1)}| \geq a_\star$, yields $\alpha_{T_2}^2 \geq 1 - \varepsilon$, where $\alpha_t := |\langle w^{(t)}, u\rangle|$ with $u = (e_1 + \text{sign}(a^{(T_1)})e_2)/\sqrt{2}$.

**Interpreting the Result**   Let

$$B_{N,d,\delta} := 1 + C\left(\sqrt{\frac{d\log(2d/\delta)}{N}} + \frac{d\log(2d/\delta)}{N}\right)$$

be the deterministic bound from Lemma 5, so that $\mathbf{Pr}[\|\widehat{M}\|_2 \leq B_{N,d,\delta}] \geq 1 - \delta$. Let $c_0 > 0$ be the universal constant from Lemma 10, chosen small enough that Lemma 1 also applies, and set $q_\star := c_0/\sqrt{N}$. Choose

$$a_\star := \min\left\{1, \frac{1}{2\eta B_{N,d,\delta}}, \frac{1}{32\eta B_{N,d,\delta}\sqrt{d}}, \sqrt{\frac{c_0}{64\, B_{N,d,\delta}\sqrt{N}\sqrt{d}}}\right\}.$$

In particular, for $d \leq N \leq d^2$,

$$a_\star = \sqrt{\frac{c_0}{64\, B_{N,d,\delta}\sqrt{N}\sqrt{d}}} = O\left(\frac{1}{(Nd)^{1/4}}\right),$$

which yields

$$T_1 \lesssim \frac{N^{1/4}}{\eta d^{1/4}}, \qquad T_2 \lesssim \frac{(Nd)^{1/4}}{\eta} \log\left(\frac{d}{\varepsilon}\right). \tag{B.14}$$

The total number of steps needed decreases as $N$ gets smaller. The gain of the two-phase schedule is that it avoids entering phase 2 with a *too small* outer gain $|a|$: since the $w$-update is scaled by $a_t$, small $|a_t|$ slows representation learning even if $w_0$ has typical random alignment. Phase 1 increases $|a_t|$ using the stronger fixed-batch bootstrap signal $|q^{(0)}| \sim 1/\sqrt{N}$, whereas the analogous population bootstrap signal at random initialization is only $|w^\top M w| = \Theta(1/d)$.

### B.5   Proof of Corollary 2: training first phase on random labels

We prove Corollary 2, which provides the convergence for a modified 2-phase training where the first phase uses random labels $y$ drawn i.i.d. uniformly on $\{-1, +1\}$. The proof largely follows that of Theorem 1, and we highlight modifications below.

**Phase 1: small-set training with random labels**   We state the random-label versions of the constant-probability lower bound for $|q^{(0)}|$ (Lemma 1) and the matrix Bernstein bound for $\widehat{M}$ (Lemma 5).

**Corollary 5.** *Assume $w^{(0)}$ is uniform on the unit sphere and the label $y$ for each sample is drawn uniformly from $\{-1, 1\}$. Then there exist universal constants $c_r > 0$ such that for all $N \geq 1$ and all $d \geq 3$,*

$$\mathbf{Pr}\left[|q^{(0)}| \geq \frac{c_r}{\sqrt{N}}\right] \geq p_{PZ}(c_r),$$

*where $p_{PZ}(c) := (1 - 1/\sqrt{2}) \cdot \frac{(1 - \frac{4}{3}c^2)^2}{3^8}$ as in Lemma 1.*

*Proof.* The calculation follows the same way as Lemma 1. The conditional mean and second moment are

$$\mu(w) = \mathbb{E}[Z \mid w] = \mathbb{E}\left[y\left(\sum_{i=1}^{d} w_i x_i\right)^2\right] = \mathbb{E}[y]\mathbb{E}\left[\left(\sum_{i=1}^{d} w_i x_i\right)^2\right] = 0$$

$$\mathbb{E}[Z^2 \mid w] = \mathbb{E}\left[(x^\top w)^4 \mid w\right] \geq 1$$

respectively. Therefore, we have

$$\mathbb{E}[(q^{(0)})^2 \mid w] = \frac{1}{N}\mathbb{E}[Z^2 \mid w] + \frac{1}{N^2}\sum_{i \neq j}\mathbb{E}[Z_i \mid w]\mathbb{E}[Z_j \mid w] \geq \frac{1}{N}.$$

Likewise, applying the hypercontractivity inequality and Paley-Zygmund to the nonnegative random variable $Y := (q^{(0)})^2$, we can get

$$\mathbf{Pr}\left[|q^{(0)}| \geq \frac{1}{\sqrt{2}}\frac{1}{\sqrt{N}}\right] \geq p_r$$

$\square$

**Corollary 6** (Bound for random label $\|\widehat{M}\|_2$). *For $y$ uniformly sampled from $\{-1, +1\}$, for any $\delta \in (0, 1)$, with probability at least $1 - \delta$,*

$$\|\widehat{M}\|_2 \leq O\left(\sqrt{\frac{d \log(2d/\delta)}{N}} + \frac{d \log(2d/\delta)}{N}\right)$$

*Proof.* The calculation is the same as Lemma 5, with $M := \mathbb{E}[yxx^\top] = 0$ because $y$ is uniformly random. $\square$

Replacing Lemma 1 and Lemma 5 with Corollary 5 and Corollary 6, we set $q_\star := c_r/\sqrt{N}$, and the first phase follows the true label case.

**Phase 2: population training with real labels**   We give the random label version of the simultaneous sign alignment as in Lemma 10.

**Lemma 12** (Random-label bootstrap and population sign alignment). *Suppose that in phase 1 the labels are independent random signs $\xi^{(s)} \sim Unif\{-1, +1\}$, independent of the inputs, and define the random-label empirical matrix*

$$\widetilde{M} := \frac{1}{N}\sum_{s=1}^{N} \xi^{(s)} x^{(s)} x^{(s)\top}.$$

*Let*

$$q^{(0)} := {w^{(0)}}^\top \widetilde{M}\, w^{(0)}, \quad q_{\mathrm{pop}}(w) := w^\top M w.$$

*There exist universal constants $c_r > 0$ and $p_{\text{all}}^{\text{rand}} > 0$ such that, with*

$$\mathcal{P}_0^{\text{rand}} := \left\{ \text{sign}(a^{(0)})\, q_{\text{pop}}(w^{(0)}) \geq \frac{3}{4d}, \quad \|P_{12} w^{(0)}\|_2 \leq \frac{1}{\sqrt{d}} \right\}$$

*and*

$$\mathcal{G}_{\text{sign}}^{\text{rand}} := \left\{ \text{sign}(a^{(0)}) = \text{sign}(q^{(0)}), \quad |q^{(0)}| \geq \frac{c_r}{\sqrt{N}} \right\},$$

*we have*

$$\mathbf{Pr}\left[ \mathcal{G}_{\text{sign}}^{\text{rand}} \cap \mathcal{P}_0^{\text{rand}} \right] \geq p_{\text{all}}^{\text{rand}}.$$

*Proof.* The population part follows exactly as in Lemma 10. Namely, writing

$$z_\pm := \langle w^{(0)}, u_\pm \rangle, \qquad r^2 := z_+^2 + z_-^2 = \|P_{12} w^{(0)}\|_2^2,$$

the same radial–angular argument gives a universal constant $p_{\text{pop}} > 0$ such that, with probability at least $p_{\text{pop}}$,

$$|q_{\text{pop}}(w^{(0)})| \geq \frac{3}{4d}, \qquad \|P_{12} w^{(0)}\|_2 \leq \frac{1}{\sqrt{d}}.$$

Since $a^{(0)}$ is independent of $w^{(0)}$ and symmetric about zero, with an additional probability factor $1/2$ we also have

$$\text{sign}(a^{(0)})q_{\text{pop}}(w^{(0)}) = |q_{\text{pop}}(w^{(0)})| \geq \frac{3}{4d}.$$

Thus $\mathcal{P}_0^{\text{rand}}$ holds with probability at least $p_{\text{pop}}/2$.

It remains to control the random-label empirical bootstrap. Conditional on $w^{(0)}$, for a single phase-1 sample, define

$$Z := \xi (x^\top w^{(0)})^2.$$

As in the proof of Corollary 5, the Paley–Zygmund argument applied to $(q^{(0)})^2$ gives universal constants $c_r, p_r > 0$ such that

$$\mathbf{Pr}\left[ |q^{(0)}| \geq \frac{c_r}{\sqrt{N}} \,\middle|\, w^{(0)} \right] \geq p_r.$$

Moreover, by the symmetry of the random labels, the conditional distribution of $q^{(0)}$ is symmetric about zero. Hence, conditional on $w^{(0)}$ and $a^{(0)}$,

$$\mathbf{Pr}\left[ \text{sign}(q^{(0)}) = \text{sign}(a^{(0)}), \quad |q^{(0)}| \geq \frac{c_r}{\sqrt{N}} \,\middle|\, w^{(0)}, a^{(0)} \right] \geq \frac{p_r}{2}.$$

Combining this conditional event with $\mathcal{P}_0^{\text{rand}}$ gives

$$\mathbf{Pr}\left[ \mathcal{G}_{\text{sign}}^{\text{rand}} \cap \mathcal{P}_0^{\text{rand}} \right] \geq \frac{p_{\text{pop}} p_r}{4} := p_{\text{all}}^{\text{rand}} > 0.$$

$\square$

Replacing Lemma 10 with Lemma 12, the second phase analysis follows from Lemmas 6 and 11. Choose

$$a_\star := \min\left\{ 1, \; \frac{1}{2\eta B_{N,d,\delta}}, \; \frac{1}{32\eta B_{N,d,\delta}\sqrt{d}}, \; \sqrt{\frac{c_r}{64\, B_{N,d,\delta}\sqrt{N}\sqrt{d}}} \right\},$$

where

$$B_{N,d,\delta} := C \left( \sqrt{\frac{d\log(2d/\delta)}{N}} + \frac{d\log(2d/\delta)}{N} \right)$$

in this setting. When $d < N < d^2$, $B_{N,d,\delta} \simeq \sqrt{d/N}$, $a_\star$ is upper bounded as $a_\star \lesssim \frac{1}{\sqrt{d}}$. Combining the first and second phase, we have the total number of steps bounded as

$$T(a_*) = T_1 + T_2 \lesssim \frac{a_* \sqrt{N}}{\eta} + \frac{2}{\eta a_*} \log \left( \frac{d}{\varepsilon} \right).$$

$T$ is a monotonically decreasing function in $a_\star$ when $a_\star \lesssim \frac{1}{N^{1/4}}$. Therefore,

- for $N < d^2$, $a_\star = O(\frac{1}{\sqrt{d}})$, and this gives $T = O\left( \frac{\sqrt{N}}{\eta\sqrt{d}} + \frac{\sqrt{d}}{\eta} \log\left(\frac{d}{\varepsilon}\right) \right)$.

- for $N = d^2$, $a_\star = O(\frac{1}{\sqrt{d}}) = \frac{1}{N^{1/4}}$, we obtain $T = O\left( \frac{N^{1/4}}{\eta} + \frac{N^{1/4}}{\eta} \log\left(\frac{d}{\varepsilon}\right) \right)$.

## C   Experiment details and additional results

### C.1   Experiment details

We report the architectures used for each task.

- *Single-Index Model (SIM)*: The link function for SIM is degree 3 Hermite polynomial and dimension $n = \{40, 50\}$. The default MLP in the experiments has 2 layers and hidden dimension 64, with ReLU activation function [8] and batch size 128 for mini-batch training.

  We train Transformers (encoder-type, i.e. no causal masking) with 2 layers and 4 heads and embedding dimension 64 with the fixed batch size 128, a simple 2-phase repeat can accelerate training.

- *Sparse parity*: MLP experiments default to 2-layer MLP with hidden dimension 64 and ReLU activation. For Figure 4, each heatmap was created with roughly 42 million training runs done on a single A100 in four hours.

  Transformer experiments are using encoder-only structure (i.e., without causal masking) to preserve a permutation-invariant structure of the parity task. The default Transformer has 2 layers, dimension 256, and 8 heads.

- *Modular addition*: Modular addition runs use a 4-phase training strategy, where the dataset size increases in each phase. Experiments are performed with 2-layer decoder-only Transformers (i.e., with causal masking).

- *In-context linear regression*: We use Transformers 2 layers and 4 heads and embedding dimension 64 on the in-context linear regression task with the number of context examples $k = 15$ and dimension $n = 4$. During training, the loss is computed on the last token.

**Hyperparameters**   We sweep the hyperparameters separately for each task and training setup. We tune the learning rate for SGD, and tune both the learning rate and $\beta_2$ for AdamW, with a fixed $\beta_1 = 0.9$. Learning rates are swept at multiplicative intervals of no wider than 1.2x. We consider $\beta_2 \in \{0.8, 0.9, 0.95, 0.999\}$. For mini-batch training, MLP experiments use batch size 128, and Transformer experiments use batch size 32, unless specified. Parity and SIM experiments use a test set of size 4096 and 5000 respectively.

For model initialization, all weights are initialized with Pytorch defaults. In particular, linear weights are initialized as $W_{ij} \sim \mathrm{Unif}[-1/\sqrt{d_{\mathrm{in}}}, 1/\sqrt{d_{\mathrm{in}}}]$. For MLP, we additionally experiment with Gaussian initialization (i.e., $W_{ij} \sim \mathcal{N}(0, 1/\sqrt{d_{\mathrm{in}}})$) whose standard deviation differs from that of the uniform distribution differs by a factor of $\sqrt{3}$; we get the same conclusions. For Transformer, attention is computed as $a_{i,j} \propto \exp(\frac{q_i^\top k_j}{\sqrt{d}})$, where $q_i, k_j \in \mathbb{R}^d$. For experiments with RMSNorm, we use RMSNorm with a learnable scale parameter.

**Remark 7** (Learning rate for sparse parity)**.** Sparse parity has a special structure that the population gradient of the first layer weight reveals information about the support [6], hence training with the full population can in theory leverage this information and converge quickly. However, the population gradient signal is exponentially small, hence leveraging the

---

[8]GELU for SIM and sigmoid for parity also have the similar results.

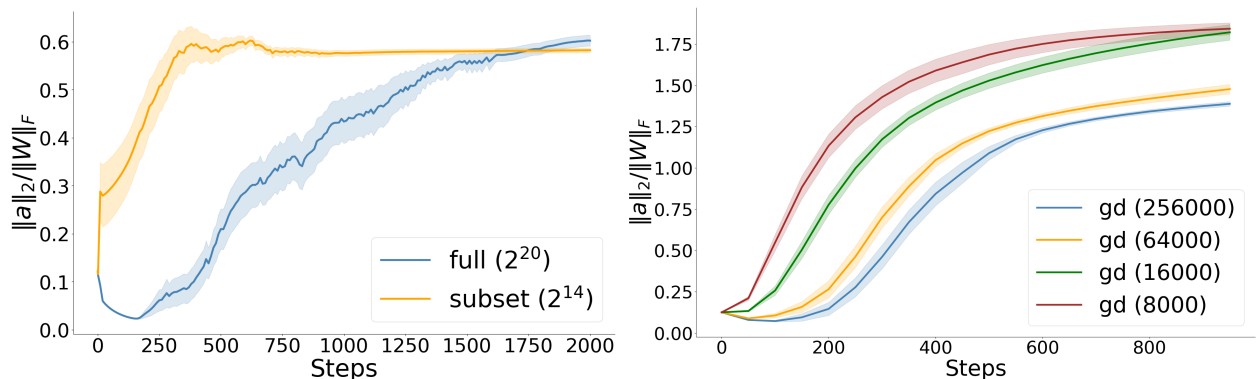

*Figure 10.* **Layer norm ratio $\|a\|_2/\|W\|_F$ increases.** Results are shown for MLP on $(20, 6)$-parity and SIM trained with gradient descent.

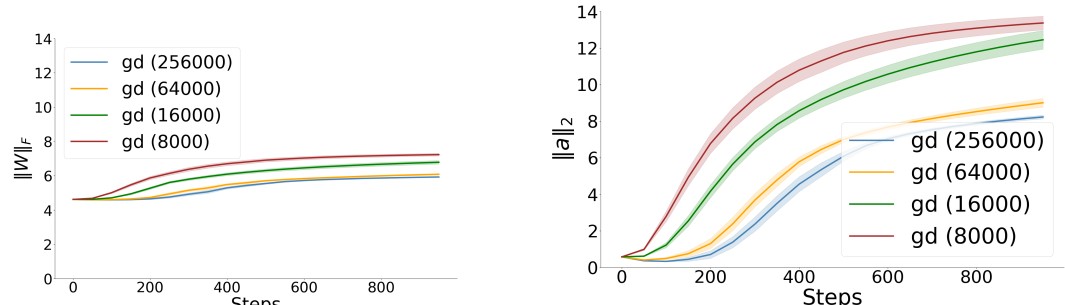

*Figure 11.* **Layer norm growth during training**. Results are shown for MLP on SIM trained with gradient descent.

signal requires an exponentially large learning rate (on the order of $d^k$) which is infeasible due to numerical limitations and movements from the second layer. Our experiments confirm this and we did not see strict speedup from increasing the learning rate.

**Data use** We provide the phase schedule used in multi-phase training.

- SIM uses a 2-phase schedule. For GD on MLP, the first phase takes 100 steps on a dataset of size 8000 and the second phase takes 900 steps on a dataset of size 64000. Ablation results with other dataset sizes are shown in Figure 18. For SGD on MLP, the first phase uses 0.01 fraction of the total amount of data seen during the online run for 100 steps, and the second phase is online training. For transformer, the first phase uses 0.005 fraction of the total amount of data seen during the online run for 800 steps, and the second phase is online training.

- Parity uses a 6-phase schedule, where each phase uses $\{0.001, 0.002, 0.005, 0.01, 0.02, 0.1\}$ of the total amount of data seen during the online run, with $\{100, 50, 20, 10, 10, 4\}$ epochs respectively. Ablation results with auto-scheduling is shown in Figure 19.

- In-context linear regression uses uses a 4-phase schedule, where each phase uses $\{0.005, 0.02, 0.05, 0.1\}$ of the total amount of data seen during the online run, with each phase running $\{1500, 1500, 1500, 10500\}$ steps respectively.

- Mod addition uses a 4-phase schedule, where each phase uses $\{0.005, 0.02, 0.05, 0.1\}$ of the total amount of data seen during the online run, with $\{20, 5, 4, 6\}$ epochs respectively.

Some of our experiments with data repetition use sampling with replacement for faster data loading, which differs from the common multi-epoch training where each epoch samples without replacement. As a remark, sampling with or without replacement correspond to different algorithms. For example, [20] showed that for linear regression, the former is equivalent (in terms of sample complexity) to gradient descent, whereas the latter is closer to online SGD which can be better

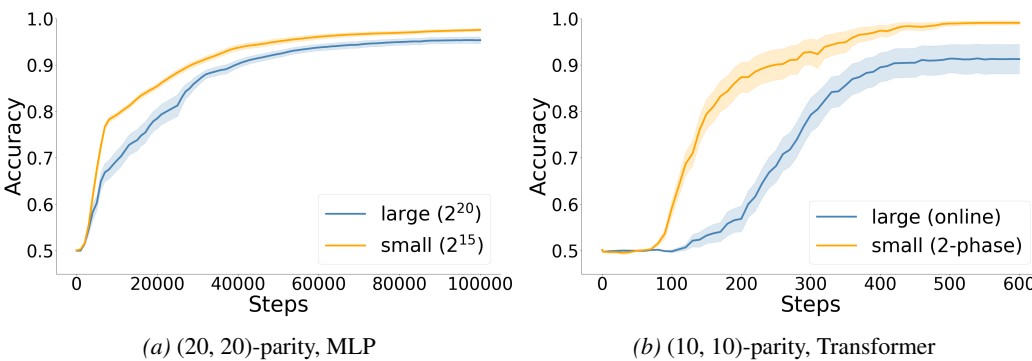

*(a)* (20, 20)-parity, MLP          *(b)* (10, 10)-parity, Transformer

*Figure 12.* **Small-vs-large gap exists for dense parity**. Results are shown for *(Left)* (20, 20)-parity with MLP and *(Right)* (10, 10)-parity with Transformer. Both are trained with full-batch gradient descent.

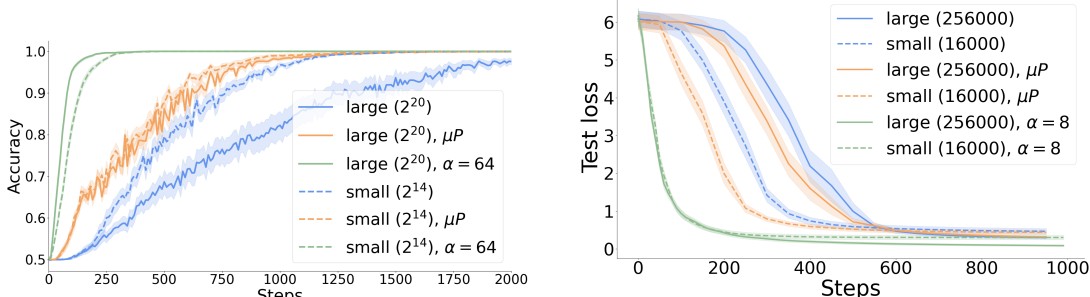

*Figure 13.* **Comparison to** $\mu P$ [30]. $\mu P$ and the $\alpha$ scaling both close the small-vs-large gap in 2-layer width-64 MLPs.

or worse than GD depending on the problem structure [26]. In our experiments though, we do not notice an empirical difference between the two based, hence we use them interchangeably.

## C.2   Additional empirical results

### C.2.1   MORE SETUPS WITH THE SMALL-VS-LARGE GAP

We report more setups where the small-vs-large gap is observed.

**Full parity**   We consider learning the full parity where $d = k$. This is a trivial task in the SQ sense and does not have a sparse structure, hence the explanations in Section 4.1 do not apply. However, the small-vs-large gap is still present, for both MLP and Transformers (Figure 12).

$\mu P$ **results**   As discussed in Section 5.2.1, a proper initialization scheme can bridge the small-vs-large gap, which prompts the question of how to choose initialization. A natural candidate is the $\mu P$ parameterization [29, 30]. As shown in Figure 13, $\mu P$ bridges the gap in 2-layer MLPs for parity but cannot close the gap for SIM. Identifying the right scheme is an interesting direction for future work.

**Transformer using full-batch updates**   The small-vs-large gap is observed on Transformers trained with full-batch updates using AdamW (Figure 14), demonstrating that the gradient variance explanation in Section 4.1 is insufficient. Due to memory constraint, we use a smaller input dimension ($d = 10$) than the MLP experiments ($d = 20$ in Figure 2).

**Transformer using mini-batch updates, with dataset input biased removed**   As discussed Section 4.1, the small-vs-large gap cannot be explained by a strong *input* bias from a smaller dataset, as the gap persists even when the input bias is removed (Figure 15a). The same conclusion holds for Transformers Figure 16.

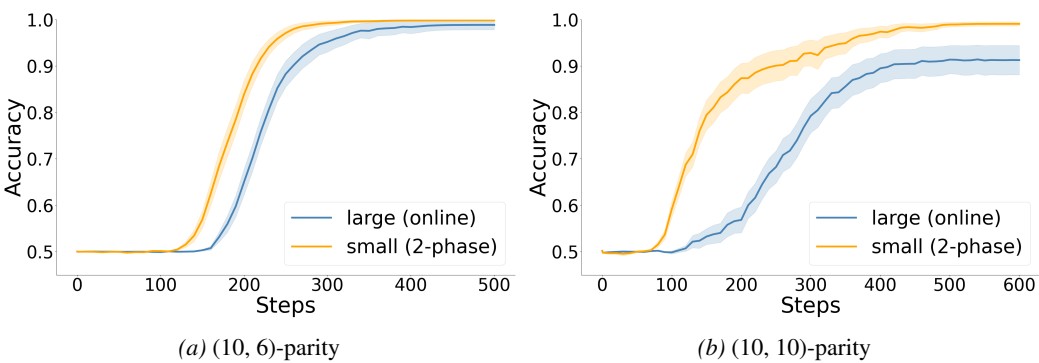

*(a)* (10, 6)-parity

*(b)* (10, 10)-parity

*Figure 14.* **Small-vs-large gap is observed in Transformer full-batch training**. Results are on (10, 6)-parity (left) and (10, 10)-parity (right).

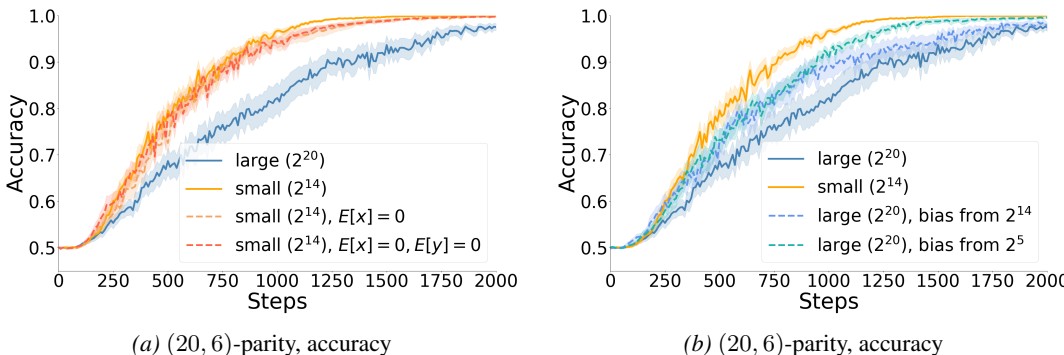

*(a)* $(20, 6)$-parity, accuracy

*(b)* $(20, 6)$-parity, accuracy

*Figure 15.* **Small-vs-large gap is not explained by input distribution biases.** *(Left)* Removing input biases does not affect the performance of training on a small set (size $2^{14}$). Removing biases means requiring $\mathbb{E}[x] = 0$, or additionally requiring $\mathbb{E}[y] = 0$ and $\mathbb{E}[x|y] = 0$. *(Right)* Introducing biases to the large set does not bridge the small-vs-large gap. The biases are taken from the empirical distribution of an size-$2^m$ set, for varying $m$. When biased with $m = 14$, large-set training still has a performance gap to training on the small set of size $2^{14}$. The maximum speedup would require $m = 5$, which is much smaller than the actual small set size and not sufficient for learning. Similar results are shown for SIM and with Transformers (Figure 16).

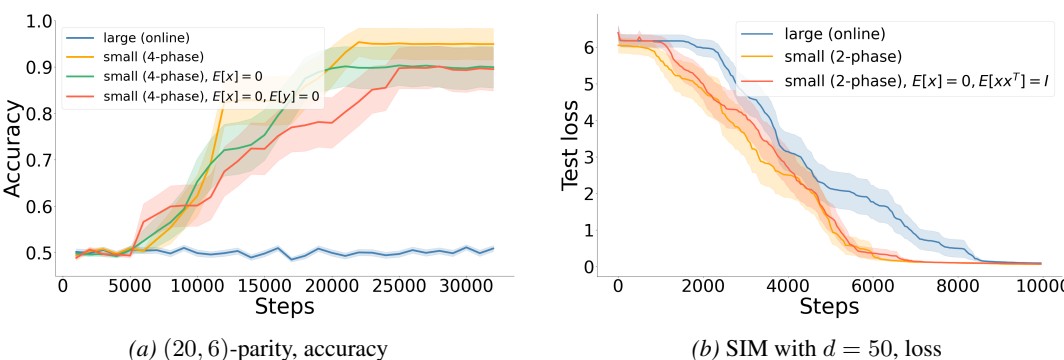

*(a)* $(20, 6)$-parity, accuracy

*(b)* SIM with $d = 50$, loss

*Figure 16.* **Repetition remains superior with dataset bias removed**. Results are based on Transformer with mini-batch updates and are consistent with the MLP results in Figure 15a.

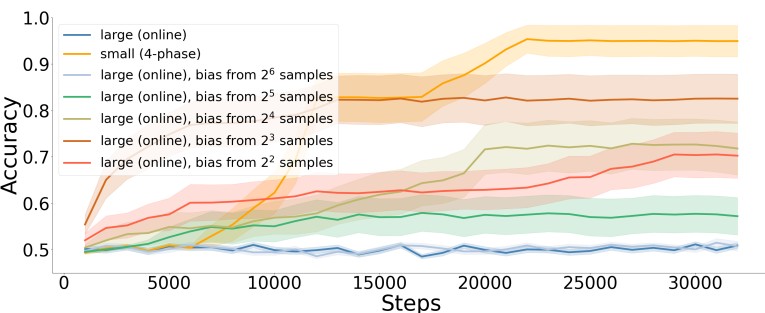

*Figure 17.* **Biasing online training does not bridge the speed gap.** Results are based on Transformer with mini-batch updates and are consistent with the MLP results (Figure 15b). For sparse parity ($d = 20, k = 6$), biasing the Bernoulli distribution with the empirical mean of $2^i$ samples (for $i \in \{2, 3, 4, 5, 6\}$) makes online training faster for certain values of $i$ (best at $i = 3$). However, to reach similar speedup as given by training on smaller datasets (marked as "4-phase"), the amount of bias required for large set (i.e., online) training would require an extremely small dataset size.

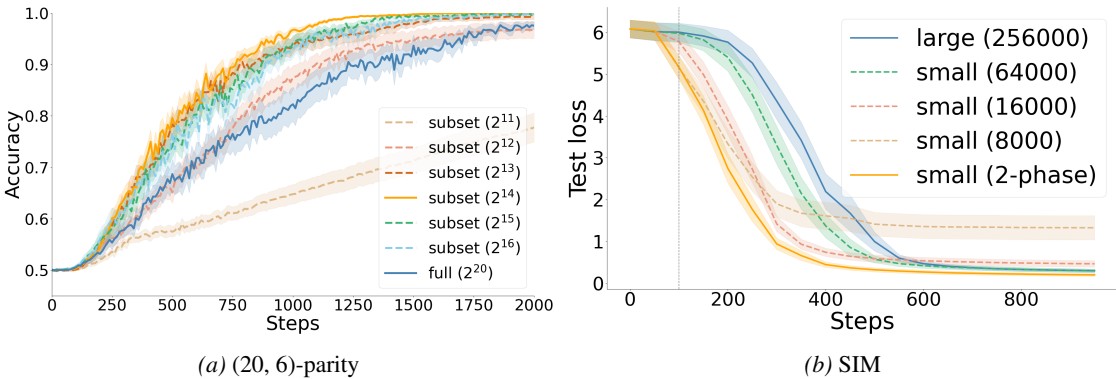

*(a)* (20, 6)-parity               *(b)* SIM

*Figure 18.* **Varying the small dataset size.** Results shown on MLP with full-batch training, for parity (left) and single-index model (right).

### C.2.2  ABLATION STUDIES

**Choosing the small dataset size**  We show ablation on the size of small-set training. A proper dataset provides learning speedup without incurring severe overfitting, as shown in Figure 18 for parity and single-index model (SIM). Note that for SIM, using a smaller dataset can lead to speedup initially but a worse loss at convergence. Hence our main results on SIM (e.g. Figure 2) adopts 2-phase training, where we first train on a small dataset (of size 8192) and then switch to a larger dataset (of size 256000). We recommend such multi-phase in general to obtain both learning speedup and generalization benefits of using the full dataset.

**Auto-scheduling for multi-phase training**  As mentioned in We also consider an alternative auto-scheduling, which the phase sizes and durations are determined automatically. Specifically, there are 6 phases, where the first and last phase is of size 1/320 and 1/50 of the amount of data seen during the online run. The intermediate dataset sizes are distributed geometrically. Each phase advances to the next one either when the training accuracy reaches 75%, or when 50 epochs have elapsed. As shown in Figure 19, this auto-scheduling achieves comparable performance to the 6-phase scheduling described above and is much faster than online training.

**Speedup from random-label training**  Our work attributes the small-vs-large gap to the layer balancing effects enabled by small-set repetitions. As discussed in Section 5.1, one strong empirical evidence for this is that training on a small set of samples with *random labels* also leads to accelerated learning. We now Figure 20 provide additional evidence on mod addition, learned with Transformers with mini-batch updates using AdamW. We train the model first on a small subset where the labels are randomly permuted, and then switch to online training for the remaining time. Specifically, the number of samples seen during the first phase is 0.5% of the second phase, repeated for 50 epochs, i.e., the random label phase takes up 20% of the total training time. As shown in Figure 20, such initial small-set random-label training speeds up

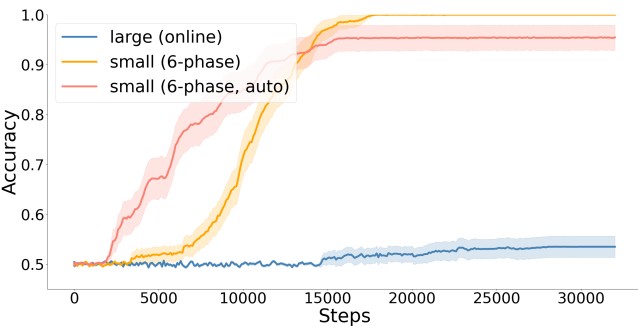

*Figure 19.* **Auto-scheduling for multi-phase learning**, where the dataset sizes across phases is distributed geometrically, and the phase duration is determined automatically based on the training accuracy. Such auto-scheduling (red) is comparable to manually selected phase scheduling (yellow), both much faster than online training.

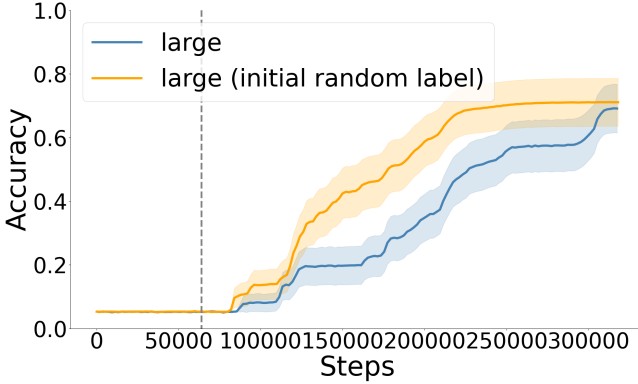

*Figure 20.* **Small-set training with random labels speeds up learning for mod addition**, complementing results in Figure 3. Compared to training directly on online batches with true labels (blue curve), adding an initial phase of repeating a small set with *random* labels help speed up learning. The gray dashed line marks the switch from training on a small set with random labels to training on online batches with true labels. Since random labels provide no learning signal, this result confirms that layer balancing is the main effect of small-set repetition. Results are shown on mod addition learned using Transformer with mini-batch updates.

learning compared to training directly with online batches with true labels.

**MLP initialization across widths**   Recall from Section 5.2.1 that proper initialization can shrink or even eliminate the small-vs-large gap. Section 5.2.1 discusses two alternative initialization schemes to the default standard initialization, namely $\mu P$ and 1-dimension simplification with an $\alpha$-scaling (i.e., dividing the first layer initialization standard deviation by $\alpha$, and multiplying the second layer's by $\alpha$). Figure 13 shows the results at width 64, where both $\mu P$ and the $\alpha$-scaling help narrow the small-vs-large gap. Figure 21 shows that for parity, $\mu P$ cannot close the gap at $m = 32$ but shows no gap for width 64 or above, which may be partly due to the effect of increasing width which reduces the gap even under the standard parameterization (Figure 9b). We hypothesize this may be because $m = 32$ is too small that it deviates too much from the infinite-width limit that $\mu P$ is designed for. However, $\mu P$ does not close the gap for SIM for the maximum width (1024) we tested (Figure 22). Further, we find that the optimal $\alpha$-scaling to stay constant across widths (Figure 23).

**Effect of Transformer QK normalization**   In Section 5.2.2, we showed that QK normalization shrinks the small-vs-large gap, for parity and SIM. Specifically, QK normalization significantly improves large-set training for both mini-batch (Figure 6c) and full-batch (Figure 27a) training. However, such improvement is not universal.

First, QK normalization worsens 4-phase training (Figure 6c). A closer investigation suggests that this is due to worse overfitting. As shown in Figure 28, training with QK normalization allows to fit the training set more quickly, while the validation accuracy remains low. Moreover, QK normalization can even worsen *online* training for some tasks, such as ICL (Figure 7d) and mod addition (Figure 27b). A mechanism understanding of these effects is left as future work.

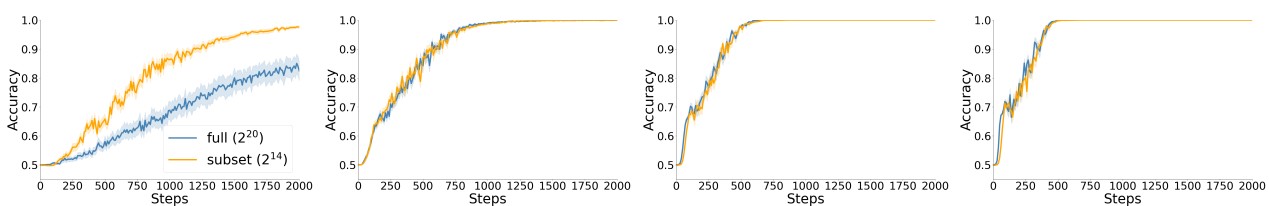

*Figure 21.* $\mu P$ **across model widths for parity.** Results are for 2-layer MLP on (20, 6)-parity trained with (full-batch) GD from $\mu P$ initialization, at various widths $m \in \{32, 64, 256, 1024\}$. $\mu P$ suffices to close the small-vs-large gap for width $\geq 64$.

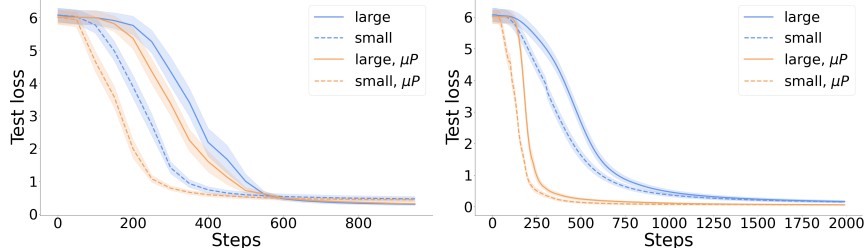

*Figure 22.* $\mu P$ **across model widths for SIM.** Results are for 2-layer MLP on SIM trained with (full-batch) GD from $\mu P$ initialization, at various widths $m \in \{64, 1024\}$. $\mu P$ doesn't close the small-vs-large gap for SIM.

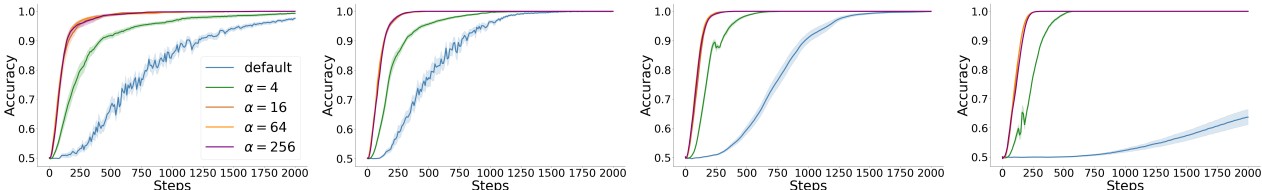

*Figure 23.* **Initialization scale holds constant across width.** Results are for MLP on (20, 6)-parity trained with (full-batch) GD on $N = 2^{14}$ samples, at various widths $m \in \{32, 64, 256, 1024\}$.

**Effect of adaptive optimizers**   We view the small-vs-large gap as related to the relative balance across layers, as supported both theoretically in Section 4.2 and empirically in Section 5. As an implication, the gap should be less pronounced when using adaptive optimizers such as AdamW, which are much less sensitive to layer scale than naive (stochastic) gradient descent. Indeed, we find that AdamW closes the gap on MLP (Figure 29), across tasks and depths. However, AdamW does not close the gap in Transformers: all Transformer experiments were conducted using AdamW and yet the small-vs-large gap persists. Hence, a full characterization of the gap is more intricate than our current explanation and likely needs to be architecture-aware.

**Which Transformer parameter benefits more from small-set training?**   We perform ablation on mini-batch training where a part of the model is updated using online data, while the rest is updated using batches repeatedly sampled from a fixed, small dataset. Figure 30 shows results for (20, 6)-parity using the default 2-layer Transformer (Appendix C). The parameter $W_v$ seems to benefit the most from small-set training, as switching to its updates to online hurts the performance the most. For $W_q, W_k$

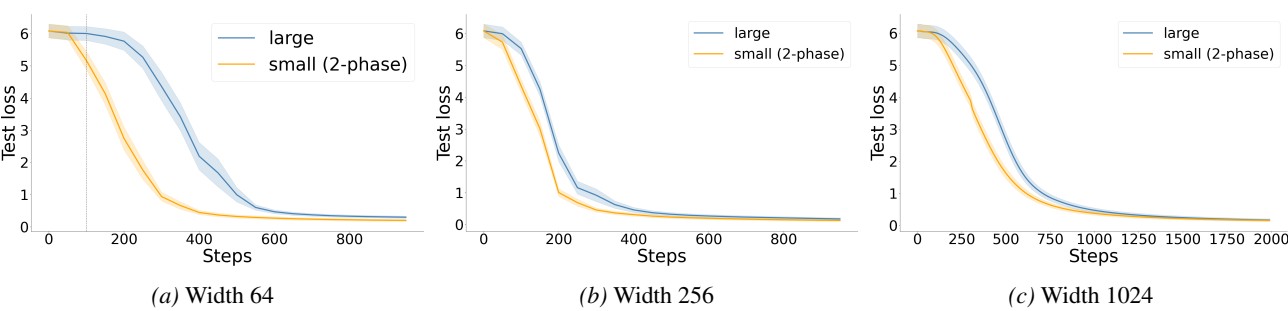

*(a)* Width 64   *(b)* Width 256   *(c)* Width 1024

*Figure 24.* **Increasing width reduces the small-vs-large gap**. Results are from 2-layer MLP with full-batch updates on SIM, where we vary the model width.

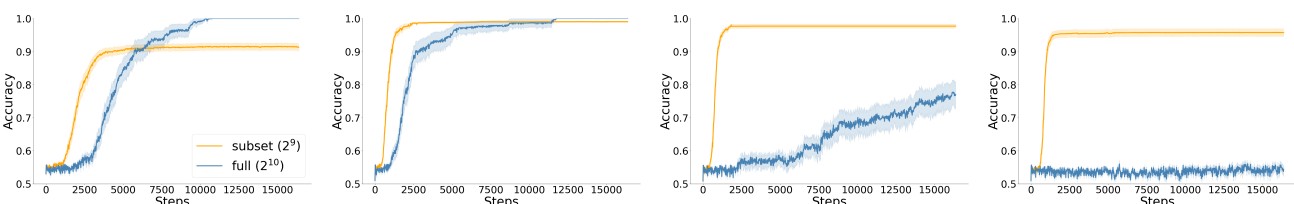

*Figure 25.* Transformer on $(10, 6)$-parity, across varying depths (2, 4, 6, 8).

*Figure 26.* **Increasing depth widens the small-vs-large gap**. Results are shown on Transformer with mini-batch Adam updates, complementing results in Figure 9a.

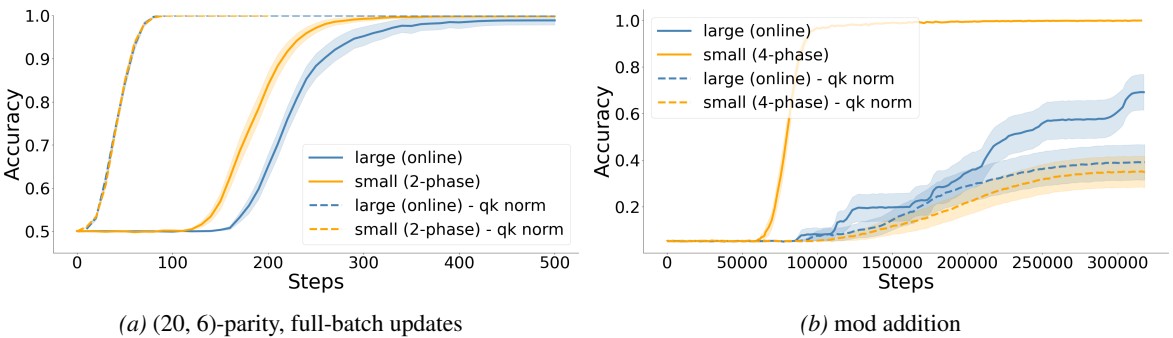

*(a)* $(20, 6)$-parity, full-batch updates   *(b)* mod addition

*Figure 27.* **Additional results on Transformer with QK normalization.** QK normalization *(Left)* removes the small-vs-large gap for parity with full-batch training, and *(Right)* worsens the training of mod addition for both online ("large") and repeated ("small") samples.

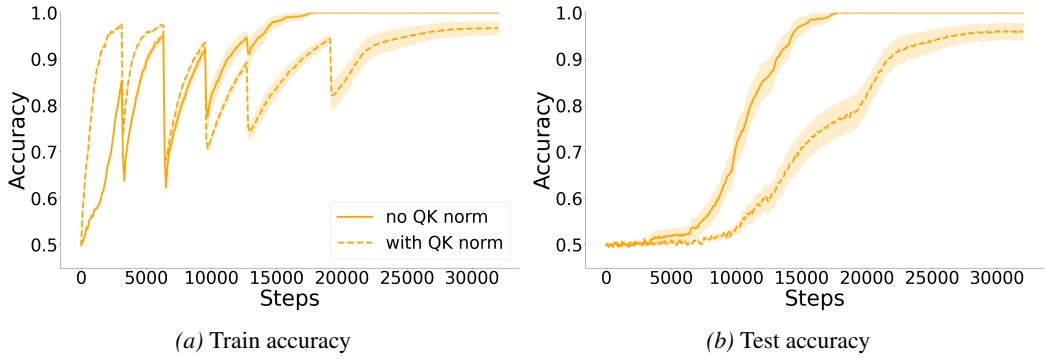

*(a)* Train accuracy   *(b)* Test accuracy

*Figure 28.* **QK slows down small-set training.** Results are shown for (20, 6)-parity with mini-batch updates. As shown in the train accuracy plot (left), QK normalization overfits to the training set quickly in the first two phases, but struggles to fit later phases where the training set sizes are larger.

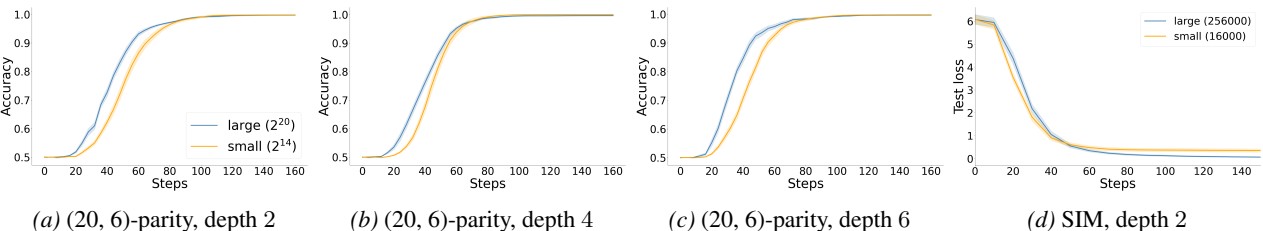

*(a)* (20, 6)-parity, depth 2     *(b)* (20, 6)-parity, depth 4     *(c)* (20, 6)-parity, depth 6     *(d)* SIM, depth 2

*Figure 29.* **Adam removes the small-vs-large gap in MLP**, across tasks and model depths. Results are shown for MLP with GD updates.

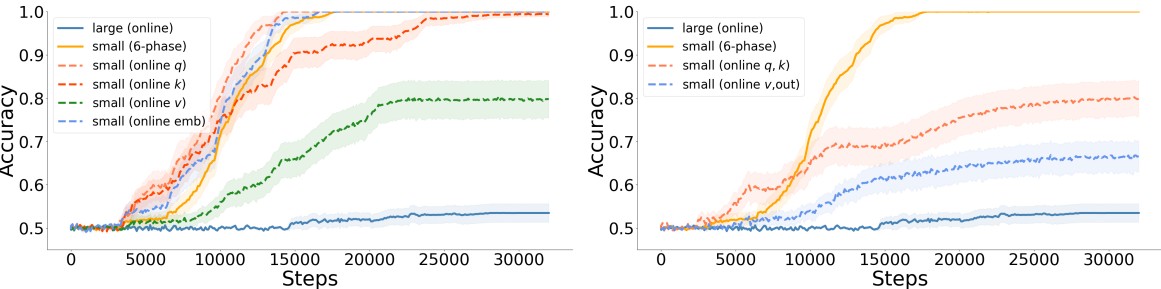

*Figure 30.* **Parameter-wise ablations of small-set training** We train a Transformer on (20, 6)-parity using 6-phase mini-batch updates, except for specific parameters which are updated using online batches. Among single parameters (*Left*), $W_v$ relies on small-set training the most, whereas the effects on $W_q, W_k$ are mild. When using online updates on a pair of parameter (*Right*), online updates on $W_q, W_k$ jointly leads to a significant slowdown.

