# OpenReview forum: "Less Data, Faster Training: repeating smaller datasets speeds up learning via sampling biases"
_ICML.cc/2026/Conference — ICML 2026 regular_

### Official Review · Reviewer_ZVcP · 2026-02-25

**Soundness:** 3
**Presentation:** 2
**Significance:** 3
**Originality:** 3
**Overall Recommendation:** 4
**Confidence:** 3

**Summary:**

This paper investigates the counterintuitive phenomenon that training on smaller datasets can lead to faster convergence in terms of compute. The authors confirm this gap exists broadly across tasks, architectures, and optimizers. They argue that prior explanations (CSQ-SQ gap, variance reduction, biased distributions) are insufficient, particularly because the gap persists under full-batch gradient descent. Instead, the paper proposes that small-dataset biases implicitly accelerate the relative norm growth between layers, effectively creating favorable layer-wise learning rates that speed up feature learning. A formal analysis on a 2-layer quadratic-activation network under 2-phase training shows that Phase 1 on a subset of size $N$ requires $O(\sqrt{N}/\eta)$ steps, and various empirical interventions (random-label pretraining, layer-wise learning rates, initialization scaling) support the hypothesis.

**Compliance With Llm Reviewing Policy:**

Affirmed.

**Key Questions For Authors:**

1. Theorem 2 analyzes a single quadratic neuron with projected gradient descent. What are the key obstacles to extending the analysis to ReLU networks or networks with width $m > 1$? In particular, does the mechanism (Phase 1 grows $a$ while $w$ stays frozen) still hold qualitatively when multiple neurons interact and break the per-neuron independence?

2. The multi-phase protocol requires specifying subset sizes and durations for each phase, yet the paper provides only high-level heuristics. How sensitive is the speedup to these choices? For example, in Figure 1, what happens if the Phase 1 subset is too small (e.g., $N = 2^6$) or the phase transition happens too early or too late? Is there a principled criterion for switching phases?

**Limitations:**

yes

**Strengths And Weaknesses:**

The paper makes a compelling empirical and conceptual contribution to understanding an important and underexplored phenomenon. The experimental coverage is impressive: the small-vs-large gap is demonstrated across four distinct tasks, two architectures, and both stochastic and deterministic optimization, with clean ablations that systematically rule out prior explanations. The proposed explanation — that small-dataset bias adjusts relative layer norms, acting as implicit layer-wise learning rates — is intuitive and well-supported by a coherent chain of evidence. The theoretical analysis, while restricted to a simplified 2-layer setting, cleanly captures the core mechanism and provides the right qualitative predictions.

The main limitation is the gap between the theoretical analysis and the empirical settings. Theorem 2 is proven for a single quadratic neuron with correlation loss and projected updates, which is a highly stylized setup that is far from the ReLU MLPs and Transformers used in experiments. The paper does not discuss whether the mechanism extends to deeper networks, different activations, or adaptive optimizers like AdamW. Relatedly, the explanation centers on 2-layer networks where feature learning occurs in the input layer, but for deeper networks and Transformers, the layer-wise learning rate story becomes considerably more complex, and the paper offers only empirical evidence without theoretical guidance. While I understand the difficulty in theoretical analysis for realistic models, a high-level explanation will still be useful.

---

> ### Author Rebuttal · Authors · 2026-03-31
>
> Thank you for the comments! We are glad that the reviewer finds our contribution compelling both empirically and conceptually.
>
> **Connection between our theoretical and empirical results**
>
> We first give an high-level overview of our theory. The population gradient of each layer depends on the product of the weights in all the other layers. The default initialization scale makes the gradient signal strength very small. The sampling bias from smaller dataset can quickly adjust the relative norm scale across layers, increasing the effective learning rate for the feature learning layer. In the following, we explain why its intuition can extend to other settings.
>
> - **Different activations**: Our theory gives the intuition that the speedup exists when the population gradient norm is very small at the initialization, whereas the subset gradient has much larger norm. This intuition applies to various activation functions. Empirically, we have verified this across ReLU, GeLU, tanh, sine, and linear activations.
> - **Adaptive optimizers e.g. AdamW**: This is an insightful question. Updates in adaptive optimizers such as AdamW are much less sensitive to gradient norm due to preconditioning. Therefore, we should expect AdamW to remove the small-vs-large gap in MLP experiments, which align closely to our theoretical setting. This is what we observe empirically, MLPs trained with AdamW show no small-vs-large gap.
>
>     However, adaptive optimizers cannot remove the small-vs-large gap universally. In particular, our Transformer results are all trained with AdamW, and the small-vs-large gap persists. While our theorem cannot directly explain this, the intuition on the norm scale and effective learning rates still holds. For example, Figure 9 shows that increasing the QK initialization scale can reduce the gap; we have additionally checked that increasing the learning rates on QK also reduces the gap.
>
> - **Deeper networks**: Increasing depth make the small-vs-large gap more pronounced, for both MLPs (Fig 11) and Transformers (Fig 12). This follows from our theory: the gradient norm of each layer depends on the product of the weights across all layers, so a suboptimal initialization scale in any layer can slow training. Searching for optimal layer-wise scales or learning rates can become increasingly infeasible with depth, whereas repeating a smaller set of data introduces a favorable optimization bias that automatically adjust the scale.
>
> For the other questions:
>
> - **Extending the theory (Q1):**
>     - *To ReLU activations*: We believe the same conclusion will hold, though the analysis will be more involved due to the more complicated gradient expression.
>     - *To including across-neuron interactions*: We believe our explanation will continue to hold for two reasons:
>         1. The correlation analysis is a good approximation of the initial training phase where the network output is small (i.e., for $f \approx 0$, $(f-y)\nabla f \approx -y \nabla f$). It is also commonly used in NN training analysis [1,2].
>         2. We empirically find the norm of $a$ to increase before that of $w$ when using the squared loss, showing a clear two-phase behavior.
> - **Sensitivity to multi-phase schedule (Q2)**: We do not yet have a formal criterion, and we may not need one: empirically, the benefits of multi-phase training are robust to hyperparameter choices (e.g., subset sizes and durations). The paper provides a high-level heuristic, which we expand below:
>     - *Dataset sizes*: we suggest 1) making the first subset relatively small so the model can both quickly reach non-trivial train set performance and deviate non-trivially from initialization; and 2) making the final subset large enough to ensure generalization. For intermediate sizes, a geometric progression works well in our experiments; e.g. for phase $i$, set $N_{i} = cN_{i-1}$ with $c>1$.
>     - *Training durations*: each intermediate phase should be long enough to reach non-trivial training set performance, but not so long that the training set is severely overfitted (e.g., having reached 0 training loss).
>
>     To make this clearer, we will update the appendix with plots showing side-by-side comparison between train and test performance curves. Specifically for the example that the reviewer mentioned, using $N=2^6$ samples will likely not improve training speed because it can be learned so quickly that the weight norms have not sufficiently grown.
>
>
> We thank the reviewer again for these insightful comments that will strengthen the paper. We will include the supporting empirical results for the above discussion in the appendix of the camera ready version.
>
> [1] Abbe et al. 23: SGD learning on neural networks: leap complexity and saddle-to-saddle dynamics.
>
> [2] Lee et al. 24: Neural network learns low-dimensional polynomials with SGD near the information-theoretic limit

---

> > ### Author Rebuttal · Reviewer_ZVcP · 2026-04-01
> >
> > My questions are fully addressed and I decided to keep my score.

---

> > > ### Author Response · Authors · 2026-04-07
> > >
> > > Thank you for confirming that your concerns have been fully resolved!
> > >
> > > If you have any further suggestions to strengthen our submission, we would greatly appreciate your feedback.

---

### Official Review · Reviewer_MwmJ · 2026-03-12

**Soundness:** 3
**Presentation:** 4
**Significance:** 3
**Originality:** 3
**Overall Recommendation:** 4
**Confidence:** 4

**Summary:**

This study explores an intriguing phenomenon in machine learning:  repeated samples from a smaller dataset often yield faster training than a larger corpus. While prior literature attributes this to CSQ vs SQ or SGD noise, the authors rule them out. Instead, the paper posits that the statistical bias of small datasets facilitates the rapid adjustment of inter-layer weight norms, effectively functioning as a dynamic layer-specific weight learning rate. Extensive experiments are conducted to validate the author's hypothesis.

**Compliance With Llm Reviewing Policy:**

Affirmed.

**Final Justification:**

My concerns were adequately addressed. I decide to keep my score 4.

**Key Questions For Authors:**

Non-mandatory:  Although the authors frankly acknowledge that data repetition may be ineffective for LLMs pre-training, I still want to see the results of this method in this scenario (with appropriate amounts of data and deep models). Besides, if we strictly perform data deduplication in the pre-training data, will this method then be effective?

**Limitations:**

Yes

**Strengths And Weaknesses:**

Strengths:

1. This is a highly analytical and thought-provoking paper that reveals the underlying mechanics of deep learning. The paper would be valuable for the community.

2. The paper challenges the status quo, identifies that the CSQ-SQ gap or gradient variance reduction fails to fully capture the phenomenon that data repetition accelerates training.

3. The empirical validations are highly creative and targeted. Multiple synthetic tasks (Sparse Parity, Single-Index Models, Modular Addition, In-context linear regression) and model architectures (MLP and Transformer) are considered.

4. The paper is well-structured. It clearly outlines the phenomenon, demonstrates why prior theories fail, establish new hypothesis, and systematically proves it with theory and experiment.



Weaknesses:

1. The results are validated only on small-scale synthetic tasks like SIM and Sparse Parity. While these tasks are widely used in the theoretical community to study the learning mechanisms of NNs, they are vastly different from real-world applications, especially for LLM pre-training, which often involves trillions of tokens.


Minors: There are some typos, e.g. reference issue in the footnote in line 163.  A missing word in “Instead, we argue that a primary to the gap is…”  in line 32.

---

> ### Author Rebuttal · Authors · 2026-03-31
>
> Thank you for the comments! We are glad that the reviewer finds our results thought-provoking and valuable for the community. We’d like to discuss how our results relate to **real-world applications** (W1, Q1), particularly LLM training.
>
> **Real-world applications (W1, Q1) and LLM training**
>
> As the reviewer noted, we do not claim that data repetition is necessarily helpful in large-scale training, due to factors such as *(near) duplications*. However, we hypothesize that repetition may be beneficial in *structured data* settings (e.g., formal reasoning). We comment on these two points below.
>
> Regarding **duplications**, we are not confident that existing deduplication methods would make the small-vs-large gap observable, since current methods primarily rely on string- or semantic-level matching, but do not adequately capture structural similarity (e.g., two code snippets can be logically identical but with different naming) which is likely an important form of duplication.
>
> For **structured data**, we are aware of the concurrent work [1] which identifies a small-vs-large gap in *LLM post-training for reasoning tasks*. Under a fixed compute budget, their results show that more epochs on fewer samples can improve the performance on AIME 24/25 and GPQA. This finding is consistent with our hypothesis that the *more structured data* amplifies the small-vs-large gap, and suggests that LLM post-training is a promising direction for further investigation. We will include this reference as supporting evidence in the camera-ready version.
>
> Thank you also for catching the typos; we will fix them in the camera ready version.
>
> [1] Kopiczko et al. 26: Data Repetition Beats Data Scaling in Long-CoT Supervised Fine-Tuning.

---

> > ### Author Rebuttal · Reviewer_MwmJ · 2026-04-02
> >
> > I read the rebuttal, which adequately addressed my concern.  I decide to keep my score.

---

> > > ### Author Response · Authors · 2026-04-07
> > >
> > > Thank you for confirming that your concerns have been fully resolved!
> > >
> > > If you have any further suggestions to strengthen our submission, we would greatly appreciate your feedback.

---

### Official Review · Reviewer_cDNh · 2026-03-13

**Soundness:** 3
**Presentation:** 2
**Significance:** 3
**Originality:** 3
**Overall Recommendation:** 4
**Confidence:** 3

**Summary:**

The authors investigate how under a fixed compute budget, training on fewer unique samples can sometimes yield faster learning and improved performance than training on a larger dataset. The authors posit that this is caused by layer wise norm scaling leads changes the layerwise learning rates. They rule out alternative explanations (SGD specific effects - SQ-CSQ)

The paper supports this hypothesis with a
1. theoretical analysis for learning 2-sparse parity using a two-layer MLP,
2. empirical evidence for across synthetic tasks for MLPs.
3. Empirical evidence for unnormalised portions of a transformer

**Compliance With Llm Reviewing Policy:**

Affirmed.

**Final Justification:**

Building on the my review. It is clear that the authors have put a great deal of work in this.

The rebuttal does address several of questions asked. However formalising them better would lead to a stronger submission. (e.g. actual experiments for Q2, making the intuitive udnerstanding concrete form Q3).

In addition the authors acknowledge and will fix several of the typographic concerns.

Given that I feel the current score is appropriate.

**Key Questions For Authors:**

1. What is the sensitivity to different initialisation methods and the geometry of the initialisation state?
2. What does this look like on real world datasets (not necessarily language corpora)? Especially datasets at scale.
What arguments support this effect persisting beyond quadratic toy models?
3. Prior work shows random-label pretraining affects learning dynamics, structural identification, and memorization. How does the current work explain the phenomenological/observations in them?
- "Pretraining with Random Noise for Uncertainty Calibration" by Cheon et. al.,.,
- in the memorization literature (e.g. "On Memorization in Diffusion Models", Gu et. al.)

**Limitations:**

yes

**Strengths And Weaknesses:**

The work is a well written highly significant paper.

The paper has a strong theoretical explanation alongside, comprehensive experiments. The experiments both confirm the pehnomeom and rule out several alternative explanations. A very large strength is they show it is robust across (synthetic) tasks and for both MLPs and transformers

It is generally well written with minor missing links. However the paper may be longer than the required page limit - the figures from the main text have been moved into the references section, making it difficult to judge the length of the main text.

Weakness :
The Code/datasets have not been released.

---

> ### Author Rebuttal · Authors · 2026-03-31
>
> Thank you for the comments, we are glad that the reviewer find our results significant and clear! We thank the reviewer for catching the typos, which we will fix in the camera ready version, together with the codebase used, including code for synthetic data generation (W1).
>
> Regarding the questions:
>
> - **Sensitivity to initialization method or geometry (Q1)**: Our results show that the small-vs-large gap can be bridged with proper initialization (e.g. Fig 5), though such initialization can be computationally expensive to find.
> In particular, our experiments show that the small-vs-large gap exists across a suite of common initialization methods, including PyTorch default initialization (uniform), Jax default initialization (normal), Kaiming initialization (with activation-dependent scaling corrections) and $\mu P$. We will include these results in the appendix.
> - **Implication on real world datasets, not necessarily languages (Q2)**: We believe that our results are more likely to transfer to scenarios with structured data (as opposed to memorization-heavy datasets) where feature learning is required. As a supporting evidence, a concurrent work (Kopiczko et al. 26) shows that more epochs on fewer samples leads to better performance under a fixed compute budget, which is an example of small-vs-large gap observed in LLM post-training. We will include this reference in the camera-ready version.
>
>     [1] Kopiczko et al. 26: Data Repetition Beats Data Scaling in Long-CoT Supervised Fine-Tuning.
>
> - **Why random labels (Sec 5.1) can provide similar acceleration (Q3)**: The intuition in our main theorem (layer norm growth) also applies to training using random label in the first phase. In our analysis, the layer norm growth comes from the $O(\sqrt N)$ sampling bias of a size-$N$ dataset, and such bias is present even if we change the label to be random. Specifically, under the same setting as Theorem 2 but with labels changed to be uniformly sampled from {-1, +1} in the first phase, we get the same convergence time of $T=\tilde{O}_d(\sqrt{N})$; the proof follows that of Theorem 2, with slight changes to Lemma 1 and Lemma 4. We will add this as a corollary to the camera-ready version, together with a discussion and the suggested references.

---

> > ### Author Rebuttal · Reviewer_cDNh · 2026-04-05
> >
> > Thanks for your work and rebuttal.
> >
> > I appreciate the clear answers to the Q1 and Q3 and look forward to the minor typographic corrections in the updated version.
> >
> > Could you provide more detail for Q2. The cited paper certainly adds some credence to the theory (wrt repetition for SFT in AIME and GPQA). What the authors might think about non quadratic models?
> >
> > Thank you

---

> > > ### Author Response · Authors · 2026-04-06
> > >
> > > Thank you for the follow-up question!
> > >
> > > Our results convey a general phenomenon: when early-stage learning is bottlenecked by weak signal, finite-sample effects from sampling biases can amplify this signal and accelerate feature learning. While we formalize this in a quadratic setting, we believe the same mechanism is more generally applicable.
> > >
> > > - Specific to activation choices, the intuition and analysis work for different activations beyond quadratic activation, including the commonly used ReLU and GeLU: our experiments use ReLU activations, and please see also our response to reviewer ZVcP (the response for ”Extending the theory (Q1)”).
> > > - In terms of domains, we expect our results on algorithmic data to transfer to reasoning tasks such as math and code. In these domains, useful gradients arise from structured patterns (e.g., algorithmic steps), but early in training these signals are often weak and noisy. Therefore, these tasks can likely benefit from small dataset’s favorable inductive biases which can help amplify certain gradient directions, for which Kopiczko et al. 26 is an example.
> > >
> > > Please let us know if there are specific points that we can expand on further, thank you!

---

### Official Review · Reviewer_hGvF · 2026-03-13

**Soundness:** 3
**Presentation:** 1
**Significance:** 2
**Originality:** 3
**Overall Recommendation:** 4
**Confidence:** 1

**Summary:**

This paper studies an inverse-scaling phenomenon under a fixed compute budget: repeatedly training on a smaller dataset can lead to faster learning than training on a larger dataset.
The authors document this small-vs-large gap across several controlled/algorithmic tasks and propose a mechanistic explanation based on bias induced by small datasets, which accelerates training via layer-wise norm growth (effectively increasing the learning rate for feature learning).
The paper further supports the hypothesis with targeted interventions that mimic the effect of small datasets by explicitly adjusting layer-wise scaling, aiming to connect the observed phenomenon to a concrete training dynamic.

**Compliance With Llm Reviewing Policy:**

Affirmed.

**Final Justification:**

This paper presents a clear and interesting claim: under a fixed compute budget, training on a smaller dataset can sometimes accelerates training on a larger one.
The proposed mechanistic explanation is thoughtful and supported by targeted interventions, but the evidence is still mostly limited to synthetic settings, which leaves its broader practical relevance unclear.
I will keep my recommendation.

**Key Questions For Authors:**

1. Realistic benchmark validation: Does the small-vs-large gap persist on at least one realistic workload under the same compute/time budget?

2. Does a carefully selected small subset (rather than a random subset) yield the same or larger training acceleration, and can the proposed norm-growth theory be extended to explain this effect?

**Limitations:**

1. Evidence is mostly from synthetic/algorithmic tasks, so generalization to realistic large-scale training remains unclear.

2. The paper needs cleaner presentation (typos/formatting).

**Strengths And Weaknesses:**

Strengths

1. Clear and focused message.
The paper communicates a crisp and somewhat counter-intuitive phenomenon: under a fixed compute budget, repeatedly training on a smaller dataset can lead to faster convergence than training on a larger dataset (“when less gives more”). The narrative is coherent, and the paper consistently ties empirical observations to a single central hypothesis.

2. Mechanistic attempt beyond observation.
Rather than only reporting the phenomenon, the paper proposes an explicit mechanism (bias-induced layer-wise norm growth / effective learning rate) and supports it with targeted interventions (e.g., manipulating layer-wise scaling), which makes the story more compelling than a purely empirical finding.

Weaknesses
1. Limited external validity due to synthetic-only evaluation.
The empirical results are largely confined to synthetic/algorithmic tasks. While this is valuable for controlled mechanistic study, it leaves open whether the phenomenon meaningfully translates to realistic modern training (e.g., vision, language modeling, instruction tuning, or fine-tuning of foundation models) where data diversity, augmentation, and optimization dynamics differ substantially.

2. Compute/acceleration framing could be strengthened.
Since the claim is fundamentally about training efficiency, the paper would benefit from more direct “time-to-target” style comparisons on realistic workloads (or at least a stronger argument that the chosen synthetic regimes faithfully capture the relevant bottlenecks). Otherwise, the work risks being interpreted primarily as an interesting controlled phenomenon rather than a broadly actionable acceleration principle.

3. Presentation/format issues.
The current manuscript has noticeable formatting problems (including typos and minor inconsistencies), which makes it harder to read and reduces confidence in polish.
These should be cleaned up in a final version.

---

> ### Author Rebuttal · Authors · 2026-03-31
>
> Thank you for the comments! We are glad that the reviewer finds our message clear and focused, and the mechanistic explanation compelling. We discuss the two main questions below.
>
> - **Implication to realistic benchmarks (W1, Q1)**: We do not claim that a clean small-vs-large gap necessarily exists in real-world datasets in general, due to two reasons:
>     - Real datasets consist of a **mixture of tasks**, and as we touch upon in line 366-384, not every task will exhibit the small-vs-large gap. We hypothesize that the small-vs-large gap requires the task to be non-convex (excluding, e.g. linear regression), and likely with a computational-statistical gap (excluding, e.g. memorization tasks).
>     - Real datasets often include **(near) duplicates** at sample or sub-sample level, which makes it hard to determine the “effective” dataset size.
>
>     Nevertheless, a concurrent work [1] reports findings consistent to ours on LLM post-training for math and coding, where they show that more epochs on fewer samples can lead to better performance under a fixed compute budget. We will include this reference as supporting evidence in the camera-ready version.
>
>     [1] Kopiczko et al. 26: Data Repetition Beats Data Scaling in Long-CoT Supervised Fine-Tuning.
>
> - **Potential actionable principle (W2)**: Following the above discussion and as noted in the paper (line 383), our findings are more relevant in settings with structured data. We believe post-training for reasoning is a promising direction, as the data is more structured compared to web-based pre-training corpora, which is also evidenced by the results in Kopiczko et al. 26.
> - **Whether data subset selection provides further benefits** **(Q2)**: This is an interesting comment: we believe that proper selected subsets may provide further acceleration over random subsets, since the two provide orthogonal benefits that likely compound positively:
>     - The benefit of a (random) small subset, as the reviewer correctly pointed out, can be explained by appropriate norm growths enabled by **sampling bias** (of magnitude $O(\sqrt{N})$). Such bias depends primarily on the *number* of samples $N$, but not *which* samples. Hence a carefully selected small dataset likely also enjoys such benefit.
>         - Related, using random labels on the small dataset which also provide speedup (see Sec 5.1, and our response to Reviewer cDNh).
>     - “Carefully selected small subsets”, such as those given by data pruning methods, are typically optimized for some notions of “quality” and can hence provide additional benefits.
>
> We also thank the reviewer for suggesting to improve the presentation and format (W3), which we will update in the camera-ready version.

---

> > ### Author Rebuttal · Reviewer_hGvF · 2026-04-03
> >
> > Honestly, I still have a few remaining questions (e.g., a concrete example of careful subset selection and more details on application to real datasets), but these are not central to the main contributions of the paper. Therefore, I will keep my score.

---

> > > ### Author Response · Authors · 2026-04-05
> > >
> > > Thank you for your follow-up and for indicating that your concerns have been resolved. We appreciate your careful reading and feedback.
> > >
> > > Regarding the two remaining points:
> > > - Subset selection: As discussed in our paper and earlier responses, our work focuses on speedups from random subset sampling, which is orthogonal and complementary to subset selection methods. We are happy to expand on specific strategies if there are particular cases you have in mind.
> > > - Applications to real datasets: We have outlined in the paper and rebuttal a representative example in reasoning-focused post-training, which is also supported by in the concurrent work of Kopickzo et al. 26.
> > >
> > > We hope this clarifies the remaining points, and we would be glad to provide further detail if helpful.

---

### Decision · Program_Chairs · 2026-04-30

**Decision:**

Accept (regular)

**Comment:**

All four reviewers recommend weak accept, and the AC concurs.

The paper presents a clear phenomenon that training on a smaller dataset can lead to faster convergence under a fixed compute budget.  It also proposes a mechanistic explanation based on sampling-bias-induced layer-wise norm growth. Reviewers found the theoretical analysis, while restricted to a stylized setting, to be coherent and well-supported by targeted empirical interventions.

The main shared concern is the restriction to synthetic tasks, and the authors acknowledge that the phenomenon may not transfer cleanly to general large-scale pretraining, for a reason: real datasets contain near-duplicates that blur the effective dataset size, and the speedup requires tasks with a computational-statistical gap that pretraining components may not satisfy. The authors argue more narrowly that the effect is most relevant for structured data, pointing to concurrent work on LLM post-training as supporting evidence. The AC finds this scoping reasonable and not damaging to the core contribution. The rebuttal was well-received, with three reviewers marking their concerns as fully resolved.